# Attention's forward pass and Frank–Wolfe

**Albert Alcalde** [* 1]   **Borjan Geshkovski** [* 2]   **Domènec Ruiz-Balet** [* 3]

## Abstract

We analyze the hardmax limit of self-attention dynamics for token embeddings in the zero-temperature regime ($\beta \to +\infty$) and relate it to finite-$\beta$ behavior. In this limit, the update rule can be viewed as a Frank–Wolfe step for a quadratic objective over the convex hull of the current token embeddings. When the key-query matrix is negative semidefinite, the dynamics converge with the standard sublinear rate $\mathcal{O}(t^{-1})$ on the quadratic energy, whereas in the positive semidefinite case, extending the hardmax rule to the convex hull induces a Voronoi structure: vertices are stationary, interior points remain in their initial cells, and each token moves along a straight line toward its cell's vertex with exponential convergence under a step-size bounded away from zero. We additionally establish well-posedness of the associated ODE limit in this regime. For finite $\beta$, we model self-attention as a Markov chain and prove *dynamic metastability*: interior tokens reach near-vertex configurations in a constant number of steps and remain trapped for times exponential in $\beta$ with high probability, before eventual collapse to some point within the initial convex hull. Thus, hardmax dynamics accurately approximate the finite-$\beta$ process over exponentially long time horizons.

## 1. Introduction

Since their introduction in the groundbreaking work (Vaswani et al., 2017), Transformers have been at the center of every major development in large language and foundation models. Various attempts have been made to analyze the inner functioning of these models. Here we focus on the question of signal propagation: given a trained Transformer and an arbitrary prompt, we study how information flows and is transformed across layers to produce the final representation. This follows a line of recent theoretical work that began with interpreting Transformers as interacting particle systems in (Lu et al., 2019; Sander et al., 2022; Geshkovski et al., 2023; 2025), and has since been developed extensively in subsequent studies.

This question is not purely theoretical: a major part of compute in large language models is expended during inference. As such, the a posteriori analysis of trained models offers a way to understand the representations they learn, with the practical aim of reverse-engineering modules of the complete architecture, so as to reduce costs. While the multi-layer perceptron (MLP) component has been optimized and parallelized through adaptive mechanisms such as mixture-of-experts (Dai et al., 2024), the attention mechanism remains a significant bottleneck due to its $O(n^2)$ complexity in the number of tokens/context length $n$. The study of signal propagation through the lens of interacting particle systems has borne fruit: some of the scaling laws derived in (Cowsik et al., 2025) have since been used in training large language models (OLMO2 7B & 13B, (Team OLMo, 2025)).

Our study of the hardmax limit reveals a simple geometric structure underlying self-attention:

- token embeddings move toward extreme points of their convex hull via a linear optimization step;

- depending on the sign of the key-query weight, this yields either clustering or a stable cell-wise partition with stationary vertices;

- at finite temperature, this structure persists over exponentially long time horizons before eventual collapse.

### 1.1. Setup

We consider encoder-only Transformers with a single head and without MLP components. Given a sequence of token embeddings $(x_i^0)_{i \in [\![1,n]\!]} \in (\mathbb{R}^d)^n$, the $(t+1)$-th layer of the architecture is given by

$$x_i^{t+1} = x_i^t + V^t \sum_{j=1}^n \frac{e^{\beta \langle B^t x_i^t, x_j^t \rangle}}{\sum_{k=1}^n e^{\beta \langle B^t x_i^t, x_k^t \rangle}} x_j^t, \qquad (1)$$

[1]Department of Mathematics, Friedrich-Alexander-Universität, Erlangen, Germany [2]Inria & Sorbonne Université, Paris, France [3]Universitat de Barcelona, Barcelona, Spain. Correspondence to: Borjan Geshkovski <borjan.geshkovski@inria.fr>.

*Proceedings of the 43rd International Conference on Machine Learning*, Seoul, South Korea. PMLR 306, 2026. Copyright 2026 by the author(s).

for all $i \in [\![1, n]\!] := \{1, \ldots, n\}$, where $V^t$ and $B^t$ are square parameter matrices, and $\beta > 0$ is fixed. In the literature (Sander et al., 2022; Geshkovski et al., 2025; 2023), (1) is referred to as the *self-attention* model. In practical implementations, the matrix $B^t$ is typically parameterized as $(Q^t)^\top K^t$, where $K^t$ and $Q^t$ are (possibly low-rank) matrices referred to as the *key* and *query* matrices, respectively—in this regard, $V^t$ is called the *value* matrix. We shall therefore call $B^t$ the *key-query matrix*.

Due to the sign of the eigenvalues of the value matrix $V^t$, the iteration (1) may diverge exponentially to $\pm\infty$. In practice, tokens are therefore renormalized at each step via *layer normalization*. Following (Geshkovski et al., 2023), we replace it by the analytically convenient renormalization $x_i^{t+1} \leftarrow (\mathsf{R}^t)^{-1} x_i^{t+1} = (I_d + V^t)^{-1} x_i^{t+1}$, which preserves the qualitative dynamics. Then, the renormalized dynamics read

$$x_i^{t+1} = (\mathsf{R}^t)^{-1} x_i^t + (\mathsf{R}^t)^{-1} V^t \sum_{j=1}^{n} \frac{e^{\beta\langle B^t x_i^t, x_j^t\rangle}}{\sum_{k=1}^{n} e^{\beta\langle B^t x_i^t, x_k^t\rangle}} x_j^t. \quad (2)$$

Throughout the paper, we assume $I_d + V^t$ is invertible.

### 1.2. Contributions and outline

The principal objective of the present paper is to study the behavior of token embeddings $x_i^t$—henceforth referred to as *particles*—following the equation obtained by taking the formal singular limit $\beta \to +\infty$ (keeping $d, n$ fixed), and to relate this knowledge to the case $\beta < +\infty$. The limit equation reads

$$x_i^{t+1} = x_i^t + (I_d + V^t)^{-1} V^t \left( \frac{1}{\#\mathscr{C}_i^t} \sum_{y \in \mathscr{C}_i^t} y - x_i^t \right), \quad (3)$$

where we denote $X^t := \{x_j^t\}_{j \in [\![1, n]\!]}$ and

$$\mathscr{C}_i^t := \left\{ y \in X^t : \langle B^t x_i^t, y\rangle = \max_{z \in X^t} \langle B^t x_i^t, z\rangle \right\}.$$

We organize the paper as follows.

- In **Section 2**, after showing that $\mathscr{C}_i^t$ is generically a singleton, we view (3) as a Frank–Wolfe update for a quadratic objective when $B^t$ is symmetric. Recall that in the setup of a convex function $\mathsf{J} : \mathcal{K} \to \mathbb{R}$ on a compact convex set $\mathcal{K} \subset \mathbb{R}^d$, the Frank–Wolfe method (Frank & Wolfe, 1956), (Bach, 2024, Chapter 9) with step-size $\gamma^t \in (0, 1)$ minimizes $\mathsf{J}$ by using a linear oracle as

$$z^{t+1} = (1 - \gamma^t) z^t + \gamma^t \arg\min_{y \in \mathcal{K}} \langle \nabla \mathsf{J}(z^t), y\rangle.$$

We also discuss an interpretation of the matrix multiplier in (3) as a pre-conditioner (Section 2.2), since throughout the subsequent analysis, we will exclusively focus on the case $V^t = h^t I_d$ with $h^t > 0$. This yields

$$x_i^{t+1} = x_i^t + \gamma^t \left( \arg\max_{y \in \mathcal{K}^t} \langle B^t x_i^t, y\rangle - x_i^t \right) \quad (\text{SA}_\infty)$$

where $\gamma^t = h^t/(1 + h^t)$ and $\mathcal{K}^t := \text{conv}\{x_i^t\}_{i \in [\![1, n]\!]}$.

- In **Section 3**, we focus on $-B^t \succcurlyeq 0$ in $(\text{SA}_\infty)$. We recover and extend results on Frank–Wolfe for convex objectives (Frank & Wolfe, 1956; Jaggi, 2013; Bach, 2024), obtaining convergence to a single cluster at the origin with a sublinear rate (*Theorem 3.1*).

- In **Section 4**, for $B^t \equiv B \succcurlyeq 0$, $(\text{SA}_\infty)$ is a Frank–Wolfe update for a concave objective. Since existing theory only provides coarse bounds on the duality gap, we instead focus on an *ad hoc* analysis. We derive an explicit geometric description: extending $\mathscr{C}_i^t$ to the convex hull induces a Voronoi tessellation in which vertices are stationary and interior particles converge along straight lines to their cell vertices, yielding exponential convergence (*Theorem 4.2*). The same arguments establish well-posedness of the associated ODE (*Theorem 4.9*), partially resolving an open question from (Geshkovski et al., 2025).

- In **Section 5**, we relate these insights to the case $\beta < +\infty$. We focus on $B^t \equiv I_d$ and $\gamma^t \equiv \gamma$ to avoid additional technicalities. Motivated by the Gumbel trick[1], we view the self-attention model as a Markov chain with transition probabilities given by the attention scores. We show that this process exhibits *dynamic metastability*: with high probability, particles rapidly approach near-vertex configurations (*Theorem 5.2*) and remain trapped for times exponential in $\beta$ (*Theorem 5.4*), after which they eventually collapse to a single cluster (*Proposition 5.1*). Thus, $(\text{SA}_\infty)$ can be seen as a valid approximation of (1) up to $O(e^\beta)$ steps.

- In **Section 6**, we complement the theory with low-dimensional experiments showing that the same qualitative picture persists beyond the exactly analyzable setting: the cell geometry may deform, but the progression from vertex clustering to metastability and eventual collapse remains visible.

We defer a comparison with prior work to **Appendix A**.

---

[1] https://francisbach.com/the-gumbel-trick/

## 1.3. Notation

We denote by $\|x\|$ the Euclidean norm of $x \in \mathbb{R}^d$, by $\langle x, y \rangle = x^\top y$ the inner product of $x$ and $y$, by $\|x\|_A := \sqrt{\langle Ax, x \rangle}$ the $A$-norm of $x$ whenever $A \succ 0$, by $B(0, r)$ the closed Euclidean ball centered at $0$ with radius $r$, and $B_1 = B(0, 1)$. For a bounded $\mathcal{K} \subset \mathbb{R}^d$, we always denote $\mathsf{d}(\mathcal{K}) = \operatorname{diam}(\mathcal{K})$.

## 2. Derivations

In this section, we further rewrite (3) and then motivate the choice of $V^t = h^t I_d$ by viewing it as a pre-conditioner.

### 2.1. The argmax is a singleton

We begin with the following lemma.

**Lemma 2.1** ($\arg\max$ is a singleton). *Suppose $B^t$ is invertible for all $t \geq 0$. Then, for almost every initial configuration $(x_i^0)_{i \in [\![1, n]\!]} \in (\mathbb{R}^d)^n$,*

$$\#\mathscr{C}_i^t = 1$$

*for all $t \geq 0$ and $i \in [\![1, n]\!]$.*

The proof can be found in **Appendix B.1**.

### 2.2. The value matrix as a pre-conditioner

By Lemma 2.1, for almost every initial configuration the hardmax dynamics (3) can be written as

$$x_i^{t+1} = x_i^t + (\mathsf{R}^t)^{-1} V^t \left( \arg\max_{y \in X^t} \langle B^t x_i^t, y \rangle - x_i^t \right). \quad (4)$$

Since linear functionals attain their maxima at extreme points, the $\arg\max$ may equivalently be taken over $\mathcal{K}^t$. When $B^t$ is symmetric, we can view (3) (for each $i$) as a Frank–Wolfe update with "matrix-valued step-sizes" for a quadratic function $\mathsf{J}(x) = \frac{1}{2}\langle B^t x, x \rangle$ over the convex set $\mathcal{K}^t$.

Throughout the rest of the paper, we focus on the case $V^t = h^t I_d$ with $h^t \geq 0$, as it is not clear how to extend our methods to the case of such "matrix-valued step-sizes". This restriction is consistent with empirical evidence that value matrices in pretrained transformers often have strong diagonal structure, and that identity-like value matrices can still yield stable training (Trockman & Kolter, 2023). We nonetheless discuss a possible interpretation of the role of the matrix $V^t$ which also motivates our particular choice. This informal discussion is almost entirely motivated by the elementary observation that for an invertible matrix $P$,

$$\arg\max_{y \in \mathcal{K}^t} \langle B^t x_i^t, y \rangle = P^{-1} \arg\max_{z \in P\mathcal{K}^t} \langle (P^{-1})^\top B^t x_i^t, z \rangle.$$

Hence the update (3), rearranged as

$$x_i^{t+1} = x_i^t + P^t \left( \arg\max_{y \in \mathcal{K}^t} \langle B^t x_i^t, y \rangle - x_i^t \right),$$

where $P^t := (I_d + V^t)^{-1} V^t$ can be understood as a *preconditioned Frank–Wolfe* iteration. The term inside the parentheses plays—as usual—the role of a direction selected by a linear oracle, and the matrix $P^t$ determines how this direction is scaled and warped. Since $\lim_{V^t \to 0} P^t = 0$ and $\lim_{V^t \to +\infty} P^t = I_d$, $P^t$ interpolates between no update and a full step depending on the magnitude and spectrum of $V^t$. Furthermore,

1. When $V^t \succ 0$, $(I_d + V^t)^{-1} V^t$ can be seen as a surrogate for a *natural gradient step*. In particular, if $V^t \approx \nabla^2 \mathsf{J}(x_i^t)$, then $P^t$ plays a role of $\left(I_d + \nabla^2 \mathsf{J}(x_i^t)\right)^{-1} \nabla^2 \mathsf{J}(x_i^t)$—a damped Newton-like correction.

2. The update can also be interpreted in the framework of *mirror descent* with a quadratic mirror map $\phi(x) = \frac{1}{2}\langle (I_d + V^t)x, x \rangle$. In this case, the matrix $P^t$ arises from mapping a dual-space step back to the primal space via the inverse Hessian $\nabla^2 \phi(x)^{-1} = (I_d + V^t)^{-1}$, followed by applying $V^t$. Thus, $P^t$ captures how the dual geometry modifies the primal update direction.

### 2.3. Shrinkage

In view of the above discussion, we now study ($\mathrm{SA}_\infty$). By definition of ($\mathrm{SA}_\infty$), we immediately deduce the following.

**Lemma 2.2** (The convex hull shrinks). *Suppose that $\gamma^t \in (0, 1)$ for all $t \geq 0$. Then, the map $t \mapsto \mathcal{K}^t$ is decreasing: $\mathcal{K}^{t+1} \subseteq \mathcal{K}^t$ for all $t \geq 0$.*

The shrinkage of the convex hull of the particles is a property that will be of significant use in what follows, which does not hold in general for arbitrary value matrices.

## 3. Negative-definite key-query

We first consider ($\mathrm{SA}_\infty$) with a symmetric $B^t$, which we reparametrize as

$$B^t = -B_*^t.$$

This allows us to rewrite ($\mathrm{SA}_\infty$) equivalently as

$$x_i^{t+1} = x_i^t + \gamma^t \left( \arg\min_{y \in \mathcal{K}^t} \langle B_*^t x_i^t, y \rangle - x_i^t \right). \quad (5)$$

Consider

$$\mathsf{J}^t(x) := \frac{1}{2}\langle B_*^t x, x \rangle,$$

which is convex when $B_*^t \succcurlyeq 0$, and $\nabla \mathsf{J}^t(x) = B_*^t x$. Thus (5) is a standard Frank–Wolfe scheme for $\mathsf{J}^t$ over the convex set $\mathcal{K}^t$. Adapting mostly standard theory (Bach, 2024, Chapter 9.3) to $\mathsf{J}^t$, we can show the following.

**Theorem 3.1** (Frank–Wolfe convergence (to a cluster)). *Suppose $B_*^t - B_*^{t+1} \succcurlyeq 0$ and $B_*^t \succcurlyeq 0$ for all $t \geq 0$. Fix $\gamma^t = 2/(t+2)$, let $x^\star \in \arg\min_{x \in \mathcal{K}^0} \frac{1}{2}\langle B_*^0 x, x \rangle$. Then for all $i \in [\![1, n]\!]$, particles evolving according to (5) satisfy*

$$\mathsf{J}^t(x_i^{t+1}) - \mathsf{J}^t(x^\star) \ \leq \ \frac{2}{t+1} \, \lambda_{\max}(B_*^0) \, \mathsf{d}(\mathcal{K}^0)^2.$$

The proof follows by adapting standard arguments (Bach, 2024, Chapter 9.3) and can be found in **Appendix B.2**. We fix $\gamma^t = 2/(t+2)$ to obtain a sublinear $\mathcal{O}(t^{-1})$ convergence rate on the quadratic energy. But in fact all the known theory (Bach, 2024, Chapter 9.3) adapts to this setting, and one can readily deduce a qualitative convergence result assuming only that $\gamma^t \in (0,1)$ satisfies $\sum_{t=0}^{+\infty}(\gamma^t)^2 < +\infty$ and $\sum_{t=0}^{+\infty} \gamma^t = +\infty$.

# 4. Positive-definite key-query

We now consider ($\mathrm{SA}_\infty$) where $B^t$ is positive definite. In this case, the system is a Frank–Wolfe scheme for

$$\min_{y \in \mathcal{K}^t} \mathsf{J}^t(y)$$

where $\mathsf{J}^t(y) = -\frac{1}{2}\langle B^t y, y \rangle$. In the setting where the objective function is concave, less is known about the Frank–Wolfe scheme. One naturally expects a dual behavior to that of the convex case—in this instance particles should converge to the boundary of the convex hull. A first result one can show follows directly from the literature; for instance, following (Yurtsever & Sra, 2022, Lemma 2.1)[2],

**Proposition 4.1.** *Suppose $B^t = \beta^t B$ for $B \succcurlyeq 0$, with $\beta^t/\beta^{t+1} = \gamma^t/\gamma^{t+1}$ for all $t \geq 0$. For all $i \in [\![1, n]\!]$, $t \geq 0$, particles evolving according to ($\mathrm{SA}_\infty$) satisfy*

$$\min_{\tau \in [\![1, t]\!]} \max_{y \in \mathcal{K}^\tau} \langle \nabla \mathsf{J}^\tau(x_i^\tau), x_i^\tau - y \rangle$$
$$\leq \frac{1}{t}\left( \frac{\mathsf{J}^1(x_i^1)}{\gamma^1} - \frac{\inf_{y \in \mathcal{K}^t} \mathsf{J}^t(y)}{\gamma^t} \right).$$

We omit the proof since it is a straightforward adaptation of (Yurtsever & Sra, 2022, Lemma 2.1), and serves no particular purpose in our analysis. The result is also not particularly informative as there is no effective control over the step $\tau$. We instead focus on making more structural assumptions on the initial configuration under which we can establish a significantly stronger result.

---

[2]Another related work is (Lacoste-Julien, 2016), which shows convergence of the same quantity with a $\mathcal{O}(1/\sqrt{t})$-rate, for general non-convex objectives.

We recall that for a convex polytope $\mathcal{K} \subset \mathbb{R}^d$ with vertices $v = (v_1, \ldots, v_\kappa)$, and a square matrix $B$, the cells are defined by

$$\mathscr{C}_i(v) \coloneqq \left\{ x \in \mathcal{K} \colon \langle Bx, v_i \rangle = \max_{y \in \mathcal{K}} \langle Bx, y \rangle \right\}. \quad (6)$$

Our main result of this section is

**Theorem 4.2** (Convergence to vertices). *Let $B^t \equiv B \succ 0$ and $\gamma^t \in (0,1)$ for all $t \geq 0$. Consider an initial configuration $(x_i^0)_{i \in [\![1,n]\!]} \in (\mathbb{R}^d)^n$ such that*

*1. the vertices $v = (v_1, \ldots, v_\kappa)$ of $\mathcal{K} \coloneqq \mathrm{conv}\{x_i^0\}_{i \in [\![1,n]\!]}$ satisfy*

$$v_j \in \mathscr{C}_j(v) \setminus \bigcup_{i \neq j} \mathscr{C}_i(v); \quad (7)$$

*2. if $x_i^0$ is not a vertex, then it doesn't lie on any face of two adjacent cells.*

*Then the map $\sigma \colon [\![1, n]\!] \to [\![1, \kappa]\!]$ such that $x_i^0 \in \mathscr{C}_{\sigma(i)}(v)$, is well-defined, and particles evolving according to ($\mathrm{SA}_\infty$) satisfy*

$$x_i^t = \left( \prod_{\tau=0}^{t-1}(1-\gamma^\tau) \right) x_i^0 + \sum_{\tau=0}^{t-1} \left( \gamma^\tau \prod_{s=\tau+1}^{t-1}(1-\gamma^s) \right) v_{\sigma(i)}$$

*for all $i \in [\![1, n]\!]$.*

*In particular, $x_i^t$ converges to $v_{\sigma(i)}$ as $t \to +\infty$ at rate $\prod_{s=0}^{t-1}(1-\gamma^s)$, which is exponentially fast whenever $\inf_s \gamma^s > 0$.*

We provide the proof in **Appendix B.3**, which straightforwardly follows after studying some geometric properties of the cells defined in (6). We also comment on the possible genericity of the first condition in the statement in Remark 4.7.

*Remark* 4.3 (Time-dependent key-query). The key takeaway from the proof of Theorem 4.2 is that, due to the assumptions on the initial polytope, particles originating from the vertices remain fixed, while particles inside a cell move toward the cell's vertex via linear interpolation. This constitutes one step, and since $B^t$ (and thus the polytope $\mathcal{K}^t$) is constant, the argument can be iterated.

Extensions to time-dependent matrices $B^t \succ 0$ are possible but technically more involved. If $B^t = \beta^t B$ with $\beta^t > 0$, the inner-product ordering is preserved and the argument extends directly. More generally, as long as the ordering of $\langle B^t x, v_i \rangle$ at the initial vertices is preserved for all $x \in \mathcal{K}^t$, the cell structure remains fixed; under weaker assumptions, particles still move toward local maximizers, but the attracting vertices may vary with $t$.

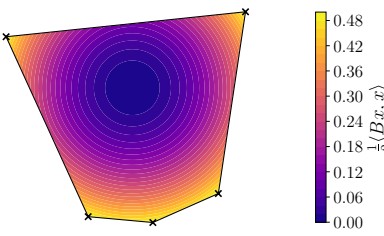

**Figure 1.** Plot of the quadratic $x \mapsto \frac{1}{2}\langle Bx, x \rangle$ with $B = \mathrm{diag}(1, 2)$, over the convex hull of 5 vertices.

## 4.1. The cells

We begin by studying the geometry of the cells $\mathscr{C}_i(v)$ in (6). Throughout this section, $\mathcal{K} \subset \mathbb{R}^d$ is a convex polytope with vertices $v = (v_1, \dots, v_\kappa)$, and $B \succ 0$.

**Lemma 4.4** (Cell geometry). *The cells $\mathscr{C}_i(v)$ satisfy:*

1. *Each cell $\mathscr{C}_i(v)$ is convex.*

2. *They have pairwise disjoint interiors:* $\mathrm{int}(\mathscr{C}_i(v) \cap \mathscr{C}_j(v)) = \varnothing$ *for $i \neq j$.*

3. *They form a partition of $\mathcal{K}$, i.e.,* $\bigcup_{i \in [\![1,\kappa]\!]} \mathscr{C}_i(v) = \mathcal{K}$.

The proof is included in **Appendix B.4**.

FURTHER OBSERVATIONS ON THE CELL GEOMETRY

One can naturally ask if the cells $\mathscr{C}_i(v)$ coincide with well-known tessellations, such as the Voronoi diagram. It turns out that this is indeed the case if the vertices all lie on the same level set of

$$\mathsf{J}(x) := \frac{1}{2}\langle Bx, x \rangle.$$

**Proposition 4.5** (Voronoi cells). *Suppose $B \succ 0$, let $v = (v_1, \dots, v_\kappa)$ be the vertices of the convex polytope $\mathcal{K} \subset \mathbb{R}^d$, and suppose that $\mathsf{J}(v_i) = c > 0$. Define the $B$-norm Voronoi cells*

$$\mathsf{Vor}_B(v_i) :=$$
$$\left\{ x \in \mathbb{R}^d : \|x - v_i\|_B \leq \|x - v_j\|_B \quad \forall j \in [\![1,\kappa]\!] \right\}.$$

*Then, for each $i \in [\![1,\kappa]\!]$,*

$$\mathscr{C}_i(v) = \mathsf{Vor}_B(v_i) \cap \mathcal{K}.$$

The proof can be found in **Appendix B.5**.

*Remark* 4.6 (Power-diagram interpretation). The assumption in Proposition 4.5 that the vertices $\{v_i\}_{i=1}^\kappa$ lie on a common level set of $\mathsf{J}(x) = \frac{1}{2}\langle Bx, x \rangle$ is not essential. In general, the comparisons $\langle Bx, v_i \rangle \geq \langle Bx, v_j \rangle$ define the intersection of $\mathcal{K}$ with the $B$–power diagram (Aurenhammer,

1987) generated by sites $v_i$ with weights $w_i = \|v_i\|_B^2$. The equal-energy assumption corresponds to uniform weights, in which case the power diagram reduces to the $B$–Voronoi diagram.

WHEN DO THE VERTICES BELONG (ONLY) TO THEIR OWN CELL?

It is not necessarily true that each vertex belongs to its own cell. This leads to the natural question of determining conditions under which each vertex belongs only to its own cell. We provide a couple of comments on this issue.

*Remark* 4.7 (Gaussian vertices in high dimension). Let $v_1, \dots, v_\kappa \overset{\text{i.i.d.}}{\sim} \mathcal{N}(0, I_d)$ and $B \succ 0$ with condition number bounded independently of $d$. Assume that $\kappa$ is fixed while $d \to +\infty$. Then, with probability tending to 1 as $d \to +\infty$, for all $i \in [\![1,\kappa]\!]$,

$$\langle Bv_i, v_i \rangle > \langle Bv_i, v_j \rangle \quad \text{for all } j \neq i,$$

that is, each vertex $v_i$ belongs to its own cell $\mathscr{C}_i(v)$ and no pair of vertices belongs to the same cell.

Indeed, since $B$ is positive definite with bounded condition number, there exist constants $0 < \lambda_{\min} \leq \lambda_{\max} < \infty$ independent of $d$ such that

$$\lambda_{\min}\|x\|^2 \leq \langle Bx, x \rangle \leq \lambda_{\max}\|x\|^2 \quad \text{for all } x \in \mathbb{R}^d.$$

For $v_i \sim \mathcal{N}(0, I_d)$, we have $\|v_i\|^2 \sim \chi_d^2$, which concentrates around $d$ with fluctuations of order $O(\sqrt{d})$, so

$$\langle Bv_i, v_i \rangle \in \left[ \lambda_{\min}d - O(\sqrt{d}), \lambda_{\max}d + O(\sqrt{d}) \right]$$

with high probability. On the other hand, for $i \neq j$, since $v_i$ and $v_j$ are independent Gaussians, $\langle Bv_i, v_j \rangle = \langle v_j, Bv_i \rangle$ is a Gaussian random variable with zero mean and variance

$$\mathbb{E}[\langle Bv_i, v_j \rangle^2] = \mathbb{E}\left[ v_j^\top Bv_i v_i^\top Bv_j \right] = \mathrm{tr}(B^2) = O(d),$$

so the typical size of $\langle Bv_i, v_j \rangle$ is of order $O(\sqrt{d})$. Thus, with high probability,

$$\langle Bv_i, v_i \rangle - \langle Bv_i, v_j \rangle \geq \lambda_{\min}d - O(\sqrt{d})$$

for all $j \neq i$. The inequality thus holds with probability tending to 1 as $d \to +\infty$.

*Remark* 4.8 (An angle condition). Suppose $B = I_d$. Fix a vertex $v_i$ and consider all adjacent vertices:

$$\mathrm{neigh}(v_i) := \big\{ v \in \{v_\ell\}_{\ell \in [\![1,\kappa]\!]} :$$
$$\text{there exists an edge connecting } v \text{ and } v_i \big\}. \quad (8)$$

We have $v_i \in \mathscr{C}_i(v)$ if and only if for every $v \in \mathrm{neigh}(v_i)$, the angle between 0 and $v$ centered at $v_i$, $\angle_{v_i}(v, 0)$, satisfies

$$\angle_{v_i}(v, 0) < \frac{\pi}{2} \quad \text{for all } v \in \mathrm{neigh}(v_i). \quad (9)$$

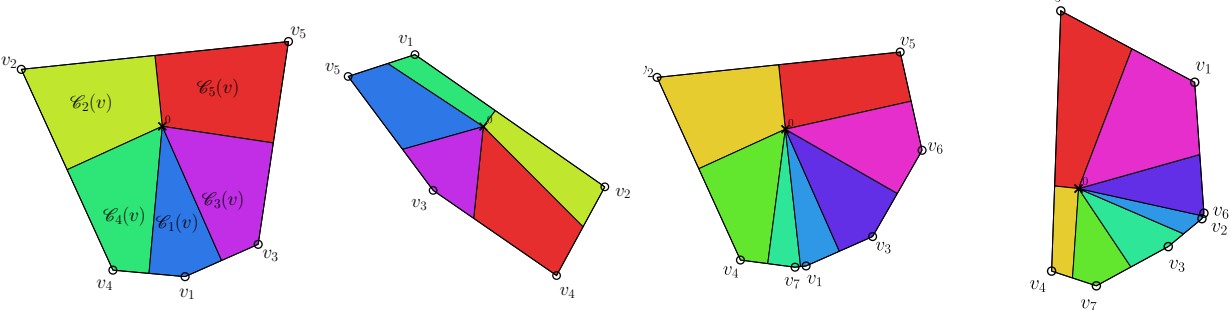

**Figure 2.** Each panel shows the cells $\mathscr{C}_i(v)$ (colored). Vertices on $\mathbb{S}^1$ yield Voronoi cells intersected with $\mathcal{K}$. Left to right: $\kappa = 5$ with vertices in $\mathbb{S}^1$, $\kappa = 5$ with arbitrary vertex norms, $\kappa = 7$ with vertices in $\mathbb{S}^1$, and $\kappa = 7$ with arbitrary vertex norms.

More generally, if $B \succ 0$, consider the level sets (ellipsoids) of J:

$$\mathscr{L}(v) := \left\{ x \in \mathbb{R}^d : \frac{1}{2}\langle Bx, x\rangle = \frac{1}{2}\langle Bv, v\rangle \right\}.$$

Let $\mathscr{H}(v)$ be the hyperplane tangent to $\mathscr{L}(v)$ at $v$, with equation $\langle a_v, x\rangle + b_v = 0$. Then, $v_i \in \mathscr{C}_i(v)$ if and only if

$$\langle a_{v_i}, v\rangle + b_{v_i} > 0 \quad \text{for all } v \in \text{neigh}(v_i) \cup \{0\},$$
$$\text{or} \tag{10}$$
$$\langle a_{v_i}, v\rangle + b_{v_i} < 0 \quad \text{for all } v \in \text{neigh}(v_i) \cup \{0\}.$$

The idea is that (10) is like being a local maximizer of J. The second part can be shown by contradiction: the hyperplane tangent to the level set at the vertex is defined by the gradient at that point. Therefore, if two neighbors lie on different sides of this hyperplane, there must exist a direction along which the value increases. When $B = I_d$, this reduces to the angle condition (9).

### 4.2. The ordinary differential equation

The considerations of Theorem 4.2, interestingly, also allow us to make sense of a continuous-time version of (4)—a first guess is

$$\dot{x}_i(t) = \underset{y \in \{x_j(t)\}_{j \in [\![1,n]\!]}}{\arg\max} \langle Bx_i(t), y\rangle - x_i(t), \quad t > 0. \tag{11}$$

This is not a trivial question at first glance, since most of the classical ODE theory (Cauchy-Lipschitz, Osgood, DiPerna-Lions...) does not apply—the right-hand side is not even continuous! One can ensure existence by looking for solutions in the class of Filippov solutions (Filippov, 1988), but uniqueness is then an arduous procedure. We instead see that—under the conditions on the initial configuration as in Theorem 4.2—well-posedness can be ensured with elementary arguments.

**Theorem 4.9.** *Let $B \succ 0$ and consider an initial configuration $(x_i^0)_{i\in[\![1,n]\!]} \in (\mathbb{R}^d)^n$ such that*

1. *the vertices $v = (v_1, \ldots, v_\kappa)$ of $\mathcal{K} := \text{conv}\{x_i^0\}_{i\in[\![1,n]\!]}$ satisfy*

$$v_j \in \mathscr{C}_j(v) \setminus \bigcup_{i \neq j} \mathscr{C}_i(v);$$

2. *if $x_i^0$ is not a vertex, then it doesn't lie on any face of two adjacent cells.*

*Then for any $T > 0$, the Cauchy problem for (11) with data $x_i(0) = x_i^0$ admits a unique solution $(x_i(t))_{i\in[\![1,n]\!]} \in C^0([0,T]; (\mathbb{R}^d)^n)$, which is continuous with respect to the initial data, and satisfies $x_i(t) \in \mathcal{K}$ for all $i \in [\![1,n]\!]$ and $t \in [0,T]$.*

The proof may be found in **Appendix B.6**.

The study of this equation is motivated by (Geshkovski et al., 2023, Section 8.1.2) (see also (Geshkovski et al., 2025, Problem 6)), where the authors argue that this singular limit is the appropriate object to describe the long-time behavior of the continuous-time self-attention dynamics (under a specific rescaling in which $\beta$ amounts to $e^{2t}$). However, they do not establish well-posedness of the equation, nor do they justify the singular limit, due to potential non-uniqueness of the $\arg\max$ in non-generic configurations (e.g., T-junctions formed by three particles).

## 5. What about (soft) attention?

The goal of this section is to transfer the results from the previous sections on the model (SA$_\infty$) to the actual self-attention dynamics (2). We focus on the case[3] $V^t = hI_d$

---

[3]Time-dependent step-sizes $\gamma^t$ could be considered under suitable decay assumptions, but are omitted to keep the arguments transparent. The restriction $B^t = I_d$ is adopted for notational convenience: the conclusions of this section extend to any $B \succ 0$ by replacing the Euclidean inner product and norm with their $B$-counterparts.

and $B^t = I_d$. The model then reads

$$x_i^{t+1} = (1 - \gamma)x_i^t + \gamma \sum_{j=1}^{n} \frac{e^{\beta \langle x_i^t, x_j^t \rangle}}{\sum_{k=1}^{n} e^{\beta \langle x_i^t, x_k^t \rangle}} x_j^t, \quad (\mathrm{SA}_\beta)$$

where we recall that $\gamma = h/(1 + h)$. It turns out that proving convergence between the two models as $\beta \to +\infty$, particularly with a quantitative rate, is rather challenging. In fact, one cannot necessarily expect such convergence to hold in general on arbitrarily long time intervals. Indeed,

**Proposition 5.1.** *Suppose $\beta > 0$. There exists some $\gamma_* \in (0, 1)$ sufficiently small such that for all $\gamma \in (0, \gamma_*)$, the following holds. For any $(x_i^0)_{i \in [\![1,n]\!]} \in (\mathbb{R}^d)^n$ there exists $x^* \in \mathrm{conv}\{x_i^0\}_{i \in [\![1,n]\!]}$ such that for every $i \in [\![1,n]\!]$, particles following ($\mathrm{SA}_\beta$) satisfy $x_i^t \to x^*$ as $t \to +\infty$.*

The proof follows mutatis mutandis from that of (Geshkovski et al., 2024b, Proposition 2.1). All results of this section hold verbatim for any $B \succ 0$, which is feasible in the single-head setting considered here. In multi-head attention, the corresponding key–query matrices are typically low-rank and therefore cannot be positive definite; extending the analysis to this degenerate case remains open.

### 5.1. A Gumbel-like trick

Instead, we proceed with a different but related idea, which is perhaps even more natural. This idea is motivated by the *Gumbel trick*, which provides a convenient method for sampling from a categorical distribution.

Concretely, let $(p_1, \ldots, p_n)$ be a categorical distribution over $[\![1, n]\!]$, so

$$p_i = \frac{e^{s_i}}{\sum_{k=1}^{n} e^{s_k}}$$

for some scores $(s_1, \ldots, s_n) \in \mathbb{R}^n$. The Gumbel trick relies on the fact that the Gumbel distribution is precisely the noise distribution for which the expected maximum of perturbed scores recovers the log-partition function. Specifically, if we draw independent Gumbel random variables $g_i \sim \mathrm{Gumbel}(0, 1)$ for each $i \in [\![1, n]\!]$, then

$$\mathbb{E}\left[ \max_{i \in [\![1,n]\!]} (s_i + g_i) \right] = \log\left( \sum_{i=1}^{n} e^{s_i} \right).$$

This trick can be used to sample from the categorical distribution by returning $\arg\max_{i \in [\![1,n]\!]} (s_i + g_i)$, since

$$\mathbb{P}\left( \arg\max_{i \in [\![1,n]\!]} (s_i + g_i) = j \right) = p_j.$$

We are impelled to consider the *self-attention process*

$\{(x_1^t, \ldots, x_n^t)\}_{t \geq 0}$ defined by

$$\mathbb{P}\left( x_i^{t+1} = (1 - \gamma)x_i^t + \gamma x_j^t \right) = \frac{e^{\beta \langle x_i^t, x_j^t \rangle}}{\sum_{k=1}^{n} e^{\beta \langle x_i^t, x_k^t \rangle}}, \quad (\mathrm{SA}_\mathbb{P})$$

with fixed initial particles $(x_1^0, \ldots, x_n^0) \in (\mathbb{R}^d)^n$. Clearly this process is a Markov chain.

Interestingly, even though ($\mathrm{SA}_\beta$) converges to a point asymptotically, for certain initial configurations, the system will remain "close" to the hardmax dynamics for a time interval that is at least exponentially large in $\beta$. This is a manifestation of the so-called *dynamic metastability* or *slow motion* (Otto & Reznikoff, 2007), previously proven for the self-attention model on the unit sphere (Geshkovski et al., 2024a; Bruno et al., 2025).

### 5.2. Dynamic metastability

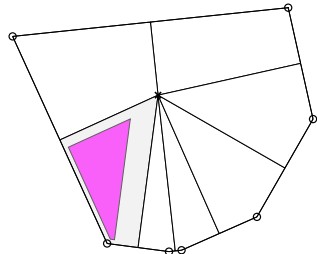

*Figure 3.* The eroded cone $\mathcal{I}_i(\eta) \ominus \delta B_1$, $\eta = 0.05$ and $\delta = 0.02$.

Given a convex polytope $\mathcal{K} \subset \mathbb{R}^d$ with vertices $v = \{v_i\}_{i \in [\![1,\kappa]\!]}$, recall the definition of the cells

$$\mathscr{C}_i(v) := \{ x \in \mathcal{K} : \langle v_i - v_j, x \rangle \geq 0 \text{ for all } j \in [\![1, \kappa]\!] \}.$$

Assuming that $v_i \in \mathscr{C}_i(v) \setminus \bigcup_{j \neq i} \mathscr{C}_j(v)$, let $\sigma : [\![1, n]\!] \to [\![1, \kappa]\!]$ be the map assigning each $x_j^0$ to its corresponding cell, i.e., $x_j^0 \in \mathscr{C}_{\sigma(j)}(v)$. Given $A \subset \mathbb{R}^d$ and $\delta > 0$, we define the interior erosion of $A$ by

$$A \ominus \delta B_1 := \{ x \in \mathbb{R}^d : x + \delta B_1 \subset A \},$$

where $B_1$ denotes the closed unit ball centered at the origin. Finally, for $\eta > 0$ we define

$$\mathcal{I}_i(\eta) := \{ x \in \mathcal{K} : \langle v_i - v_j, x \rangle \geq \eta \ \forall j \in [\![1, \kappa]\!] \setminus \{i\} \}.$$

We are now ready to state the first main result of this section.

**Theorem 5.2** (Clustering). *Let $\gamma \in (0, 1)$, and $\mathcal{K} \subset \mathbb{R}^d$ a convex polytope with vertices $v_1, \ldots, v_\kappa$ which satisfy*

i) *For any $i \in [\![1, \kappa]\!]$,*

$$\langle x - v_i, \ y - v_i \rangle > 0 \quad \text{for all } x, y \in \mathcal{K} \setminus \{v_i\}. \quad (12)$$

ii) For all $i, j \in [\![1, \kappa]\!]$,

$$\|v_i\| = \|v_j\|. \tag{13}$$

iii) There exists $c_0 > 0$ such that

$$\langle v_\iota, v_\iota \rangle \geq \langle v_\iota, v_\ell \rangle + c_0 \quad \text{if } \iota \neq \ell \in [\![1, \kappa]\!]. \tag{14}$$

Let $n \geq \kappa$. Then there exists some $\beta_* > 0$ (depending on $n$, and the geometry of $\mathcal{K}$), such that for all $\beta \geq \beta_*$, the following holds.

Consider any initial configuration $(x_i^0)_{i \in [\![1,n]\!]} \in \mathcal{K}^n$ such that

- $x_i^0 = v_i$ for $i \in [\![1, \kappa]\!]$;

- $x_i^0 \in \mathcal{I}_{\sigma(i)}(\beta^{-\frac{1}{8}}) \ominus \beta^{-\frac{1}{4}} B_1$ for $i \in [\![\kappa + 1, n]\!]$.

Then

$$\mathbb{P}\left( \bigcap_{i \in [\![1,\kappa]\!]} \left\{ x_i^{T_1} \in B(v_i, \beta^{-\frac{1}{4}}) \right\} \cap \right.$$

$$\left. \bigcap_{j \in [\![\kappa+1,n]\!]} \left\{ x_j^{T_1} \in B(v_{\sigma(j)}, C\tau) \right\} \right) \geq 1 - \beta^{-\frac{1}{8}}$$

where $C > 1$ is a universal constant,

$$\tau := \min_{i \in [\![1,\kappa]\!]} \frac{c_0}{2 \max_{j \neq i} \|v_i - v_j\|} \wedge \frac{\sqrt{2c_0}}{2}$$

$$\wedge \left( (1 - \gamma) \min_{j \in [\![\kappa+1,n]\!]} \|x_j^0 - v_{\sigma(j)}\| \right),$$

and

$$T_1 = \left\lfloor \frac{1}{\log(1-\gamma)} \log \left( \frac{\tau}{\min_{j \in [\![\kappa+1,n]\!]} \|x_j^0 - v_{\sigma(j)}\|} \right) \right\rfloor.$$

The proof can be found in **Appendix B.7**.

*Remark* 5.3 (On Theorem 5.2). We provide some comments regarding the setup of Theorem 5.2. The radius $\beta^{-1/4}$ arises from bounding the self-interaction probability (cf. Claim B.3); all other powers of $\beta$ can be improved to $\beta^{-1/4-\varepsilon}$ for any $\varepsilon > 0$, up to constants. Assumption (12) is purely technical and is only used in Claim B.2; moreover, (14) is equivalent to (7). The polynomial (rather than exponential) probability bound is due to (possibly coarse) variance estimates—see (38). Finally, the arguments extend to show the convergence of ($SA_\mathbb{P}$) toward (4) and (11) by letting $\beta \to \infty$ and $\tau \to 0$ (and rescaling time as $\gamma \to 0$), noting that the associated transient time diverges as $\tau \to 0$.

After Theorem 5.2, we enter the second phase which is summarized in the following theorem.

**Theorem 5.4** (Metastability). *Consider the setup of $\mathcal{K}$ as in Theorem 5.2. There exists some $\varepsilon_* > 0$ such that the following holds.*

*Consider any initial configuration $(x_i^0)_{i \in [\![1,n]\!]} \in \mathcal{K}^n$ such that*

$$x_i^0 \in B(v_{\sigma(i)}, C\tau)$$

*for all $i \in [\![1,n]\!]$. For $i \in [\![1,\kappa]\!]$, let $\mu_i$ denote the number of points in the ball around $v_i$:*

$$\mu_i := \#\{ j \in [\![1,n]\!] : x_j^0 \in B(v_i, C\tau) \}.$$

*For $i \in [\![1,\kappa]\!]$, relabel all the points $x_\ell^0$ as $x_{ji}^0$ if $x_\ell^0 \in B(v_i, C\tau)$. Then for any $\varepsilon \in (0, \varepsilon_*)$ and $\gamma \in (0,1)$ such that $\varepsilon/\gamma \geq 2\mathsf{d}(\mathcal{K})$, the random variable*

$$T_2 := \inf \left\{ t \geq 0 : x_{ji}^t \notin \operatorname{conv}\{x_{ji}^0\}_{j \in [\![1,\mu_i]\!]} + B(0,\varepsilon) \right.$$

$$\left. \text{for some } (i,j) \in [\![1,\kappa]\!] \times [\![1,\mu_i]\!] \right\}$$

*is such that for all $t > 1$,*

$$\mathbb{P}(T_2 \geq t) \geq 1 - \exp \left( \left(1 + \frac{\varepsilon}{\gamma}\right) \log \left(\frac{\gamma}{\varepsilon} t\right) + \right.$$

$$\left. \left(1 + \frac{\varepsilon}{\gamma}\right) \log n - \beta \frac{c_0}{2} \frac{\varepsilon}{\gamma} \right). \tag{15}$$

The proof can be found in **Appendix B.8**.

To ensure $\mathbb{P}(T_2 \geq t)$ is close to 1, we must have

$$\beta \gg \left(1 + \frac{\varepsilon}{\gamma}\right) \left( \log \left(\frac{\gamma}{\varepsilon} t\right) + \log n \right).$$

This implies that the metastable time horizon scales exponentially in $\beta$, namely, $t \lesssim \varepsilon \cdot \gamma^{-1} \cdot e^{c\beta}$ for some $c > 0$. In terms of steps, the total number of iterations before leaving the metastable state satisfies $t \cdot \gamma^{-1} \lesssim \gamma^{-2} \cdot e^{c\beta}$. Thus, for fixed spatial accuracy $\varepsilon$, the metastability lasts for an exponential number of time steps in $\beta$, provided the time-step $\gamma$ is sufficiently small. This reflects a trade-off: smaller $\gamma$ improves stability but slows down the effective time evolution. Since $\tau$ is fixed by the geometry of the polytope, (15) highlights how the metastability window depends on the interaction between temporal discretization and $\beta$.

## 6. Numerical experiments

We complement the analysis with low-dimensional numerical experiments illustrating the induced geometry beyond the exactly analyzable setting. The experiments show that relaxing the assumptions can deform the geometry and shorten metastability, while the progression from vertex clustering to eventual collapse remains visible. Code for reproducing our results is available at https://github.com/borjanG/2025-transformers-frank-wolfe.

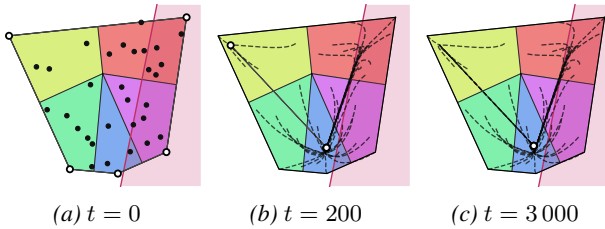

|  |  |  |
|---|---|---|
| *(a) $t = 0$* | *(b) $t = 200$* | *(c) $t = 3\,000$* |

**Figure 4.** The dynamics (2) for $\beta = 10$, $B = I_d$, and $V = 0.1I_d$, composed with residual ReLU feedforward layers. The active ReLU half-space is shaded in red.

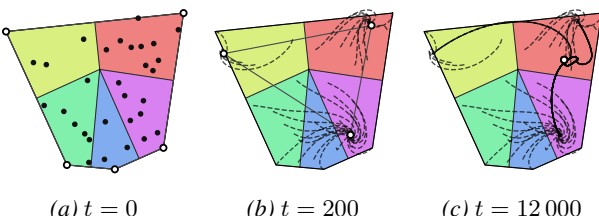

|  |  |  |
|---|---|---|
| *(a) $t = 0$* | *(b) $t = 200$* | *(c) $t = 12\,000$* |

**Figure 5.** The dynamics (2) for $\beta = 10$, $B = I_d$ and $V$ a rotation matrix by $\pi/4$.

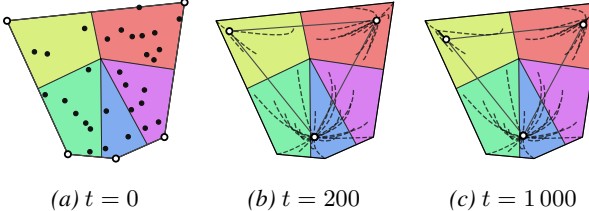

|  |  |  |
|---|---|---|
| *(a) $t = 0$* | *(b) $t = 200$* | *(c) $t = 1\,000$* |

**Figure 6.** The dynamics (2) for $\beta = 10$, $V = 0.1I_d$, and $B$ non-symmetric with skew-symmetric size $c = 0.1$.

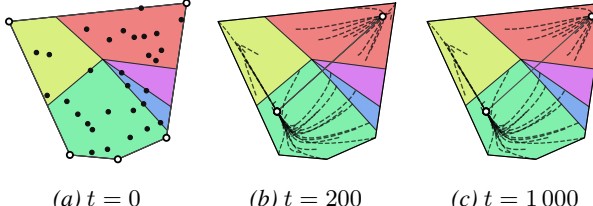

|  |  |  |
|---|---|---|
| *(a) $t = 0$* | *(b) $t = 200$* | *(c) $t = 1\,000$* |

**Figure 7.** The dynamics (2) for $\beta = 10$, $V = 0.1I_d$, and $B$ non-symmetric with skew-symmetric size $c = 1$.

**Feedforward layers** We compose each self-attention layer with a residual ReLU feedforward map

$$\mathrm{FF}(x) = x + w\,\mathrm{ReLU}(a^\top x + b),$$

with fixed $w, a \in \mathbb{R}^2$ and $b \in \mathbb{R}$. This adds a piecewise-linear deformation to the cell geometry. As shown in Figure 4, the feedforward layer changes local velocities and can accelerate merging in the active ReLU half-space. Nevertheless, the qualitative dynamics persist: particles cluster near vertices, remain metastable, and eventually collapse.

**Non-diagonal value matrices** Next, we replace the diagonal value matrix by a non-diagonal one, which can rotate or rescale the dynamics. In Figure 5, $V^t = V$ is fixed as the rotation matrix by angle $\pi/4$. The trajectories are observed to acquire a rotational component, but the qualitative behavior is unchanged: particles still cluster near vertices, remain metastable, and eventually collapse.

**Non-symmetric key-query matrices** Next, we test the robustness of our theory to non-symmetric key-query matrices. Writing $B = \mathrm{sym}(B) + \mathrm{skew}(B)$ separates the symmetric component, which is covered by our theory, from the skew-symmetric component, which induces rotations. For non-symmetric $B$, the update is no longer a gradient Frank–Wolfe scheme, but it retains a conditional-gradient-type form with direction $Bx$. We simulate the case $B = \left(\begin{smallmatrix} 1 & c \\ 0 & 1 \end{smallmatrix}\right)$. As shown in Figure 6, for $c = 0.1$ the cell geometry is only mildly deformed and metastability remains visible. The case $c = 1$ is shown in Figure 7, where rotational effects are more dominant and the cell structure is more distorted, yet the metastable phenomenon remains visible.

## 7. Concluding remarks

We analyze the hardmax self-attention dynamics (3) — the $\beta \to +\infty$ limit of softmax attention — in the regime where the key-query matrix $B^t$ is symmetric and of fixed sign. We further relate this singular-limit model to the finite-$\beta$ dynamics and establish dynamical metastability for the latter. Finally, parts of our discrete-time analysis are key in proving the well-posedness of a singular ODE that arises naturally in the asymptotic study of continuous-time self-attention.

The present analysis has some limitations: we prove metastability for constant key-query matrices $B \succ 0$ and simple value matrices. Section 6 suggests that the geometric picture is informative beyond these assumptions; extensions to this broader setting remain an open direction for future work.

## Acknowledgements

B.G. thanks Francis Bach for pointing him to the Frank–Wolfe method and the Gumbel trick. A.A acknowledges funding from the European Union (Horizon Europe MSCA project ModConFlex, 101073558); B.G from a Sorbonne Emergences grant and a gift by Google, and D.RB from "France 2030" managed by the Agence Nationale de la Recherche, under the reference ANR-23-PEIA-0004.

## Impact Statement

This paper presents work whose goal is to advance the field of Machine Learning. There are many potential societal consequences of our work, none which we feel must be specifically highlighted here.

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

# A. Discussion and related work

The $n^2$ complexity of attention and counting vertices

At each layer of a Transformer, standard soft attention as in (1) requires $O(n^2)$ operations to compute all pairwise attention scores between $n$ tokens. In contrast, our analysis hints that the mechanism may only need to identify the structural extremes of the token cloud.

Concretely, let $(x_1, \ldots, x_n) \in (\mathbb{R}^d)^n$ denote the token embeddings at a fixed layer, and let $\kappa$ be the number of vertices of their convex hull. Classical output-sensitive algorithms from computational geometry compute $\kappa$ efficiently when the ambient dimension $d$ is fixed. For instance, the gift-wrapping (Jarvis march) algorithm (Jarvis, 1973) identifies one vertex at a time in $O(n\kappa)$ time using constant memory, making it especially effective when $\kappa \ll n$. Chan's algorithm (Chan, 1996) improves this to $O(n \log \kappa)$ time with $O(n)$ space, matching known lower bounds for exact enumeration. In streaming or memory-constrained settings, the convex hull can be incrementally maintained online with expected $O(n\kappa)$ cost (Cormen et al., 2009; Preparata & Shamos, 1985). The application of such algorithms to the setting of natural language processing has already borne fruit in the past—see (Vinyals et al., 2015) for instance.

Empirically, $\kappa$ appears to grow sublinearly with $n$ and often remains in the tens even for thousands of tokens. Several studies support this observation—attention in large language models typically concentrates on a small subset of input tokens, with less than 20–30% of tokens contributing meaningfully to the output (Brahma et al., 2022), and sometimes as few as 1–2% sufficing for accurate predictions in long-context inference (Synk et al., 2025)—see also (Clark et al., 2019)—such conclusions have also been made in the context of oversmoothing or rank-collapse (Dong et al., 2021; Anagnostidis et al., 2022; Shi et al., 2022; Zhai et al., 2023; Nguyen et al., 2023; Dovonon et al., 2025; Scholkemper et al., 2025; Wu et al., 2024; Alman & Song, 2025), and attention sinks (Son et al., 2024; Gu et al., 2025; Barbero et al., 2025; Sun et al., 2026). These findings align with our geometric picture where only a small number of extreme points govern the dynamics. They also motivate algorithmic strategies that exploit sparsity, such as top-$k$ attention or token pruning, to reduce the computational burden of attention (Nawrot et al., 2025; Wang et al., 2024; Desai et al., 2025).

Self-attention dynamics

Since the clear presentation in (Sander et al., 2022), the dynamics in (1) have been studied in great detail in the mathematical literature, much as in the neural ODE literature (Esteve et al., 2020; Geshkovski & Zuazua, 2022). In (Geshkovski et al., 2023), the authors consider the continuous-time version and prove various clustering results as time tends to infinity, depending on the spectral properties of the value and key–query matrices. These results are consistent with consensus phenomena known in collective-behavior models (see (Motsch & Tadmor, 2014; Tadmor, 2021) and the references therein). The mean-field case (without rescaling) is then studied in greater depth in (Castin et al., 2025). The dynamics also bear a striking similarity to the mean-shift method (Fukunaga & Hostetler, 1975).

Our work is strongly inspired by (Geshkovski et al., 2023) and focuses on the discrete-time setting; we obtain precise rates, allow time-dependent parameters, and explain intermittent behavior. A related discrete-time work is (Alcalde et al., 2025), which assumes time-independent parameter matrices and restricts to the symmetric positive-definite key–query case, without rates.

The above-cited works omit true layer normalization. With layer normalization, the dynamics evolve on the unit sphere, as observed in (Geshkovski et al., 2025), where various clustering results are proved using synthetic gradient-flow techniques. In two dimensions, the authors originally established the result only for certain values of $\beta$; this was improved in (Criscitiello et al., 2024) and then completely resolved—and, surprisingly, generalized to a small window of negative temperatures—in (Polyanskiy et al., 2025) (see (Karagodin et al., 2026) for further discussion on normalization in self-attention dynamics).

These works have since impelled a number of refinements: dynamic metastability (Geshkovski et al., 2024a; Bruno et al., 2025); bounds on the number of clusters (Geshkovski et al., 2024c); extensions to more general parameters (Burger et al., 2025; Abella et al., 2025; Koubbi et al., 2026; Agazzi et al., 2026); additive noise (Shalova & Schlichting, 2026; Kan et al., 2026; Agazzi et al., 2026); masked attention (Karagodin et al., 2024; Wu & Varshney, 2025; Duerinckx et al., 2026); the mean-field regime (Chen et al., 2025b; Zimin et al., 2025; Bruno et al., 2026; Alcalde et al., 2026; Álvarez-López et al., 2026); mean-field control (Geshkovski et al., 2024b; Adu & Gharesifard, 2024; Biswal et al., 2025; Mehta & Meyn, 2025); LoRA-style ideas (Koubbi et al., 2024; Huan & Shun, 2025; Gang et al., 2025); and applications to operator learning (Calvello et al., 2025; Yu et al., 2024). See also (Cowsik et al., 2025; Hu et al., 2024; Viswanathan et al., 2025; Bao et al., 2024; Tomihari & Karakida, 2026) for related directions; initialization issues are studied in (Cowsik et al., 2025; Giorlandino

& Goldt, 2026; Chen et al., 2025a); and connections to clustering algorithms appear in (Clarkson et al., 2025; Zimin et al., 2025).

METASTABILITY

We use the notion of dynamical metastability as in partial differential equations such as the Allen–Cahn equation (see (Otto & Reznikoff, 2007) and references therein): the dynamics rapidly approach a nearly stationary state, remain there for a very long time, and only eventually converge to equilibrium. In this regard, there are several results for the continuous-time analogue of (1) on the unit sphere; see (Geshkovski et al., 2024a; Bruno et al., 2025). Our setting is much closer to (Geshkovski et al., 2024a), whereas (Bruno et al., 2025) does not consider the limit $\beta \to +\infty$ and instead performs a perturbative analysis around the unstable equilibrium in a mean-field regime $n \to +\infty$. Beyond the substantive differences that we work in discrete time and use a different normalization mechanism, the metastable state in our setting is characterized by the hardmax dynamics.

Finally, for Markov chains similar to ($\mathrm{SA}_\mathbb{P}$), several works obtain analogous results and provide general criteria under which they hold (Gayrard et al., 2004; Bovier et al., 2005; Bovier & Den Hollander, 2016; Landim et al., 2023). Applications include models from statistical physics such as the Curie–Weiss model (Schlichting & Slowik, 2019). We leave it to future work to determine whether our model fits within these frameworks.

# B. Proofs

## B.1. Proof of Lemma 2.1

Fix $t \geq 0$, let $\mathcal{K}^t := \mathrm{conv}\{x_j^t\}_{j \in [\![1,n]\!]}$ with $v_1^t, \ldots, v_\kappa^t$ its vertices for $\kappa \leq n$. Consider

$$H_{ij}^t := \left\{ x \in \mathbb{R}^d : \langle B^t x, v_i^t - v_j^t \rangle = 0 \right\}$$

for $i, j \in [\![1, \kappa]\!]$. By construction, each $H_{ij}^t$ is a $(d-1)$-dimensional hyperplane and since $B^t$ is invertible, they have zero measure in $\mathbb{R}^d$. Since

$$H^t := \bigcup_{i,j \in [\![1,\kappa]\!]} H_{ij}^t$$

is a finite union of zero measure sets, for almost every $x \in \mathbb{R}^d$, $x \notin H^t$. Thus, the map $T^t : \mathbb{R}^d \to \mathbb{R}^d$ such that

$$T^t(x) = x + (I_d + V^t)^{-1} V^t \left( \arg\max_{y \in \mathcal{K}^t} \langle B^t x, y \rangle - x \right)$$

is uniquely defined. To extend the argument for all $t \geq 1$, we need to verify that $T^t$ does not map positive measure sets into zero measure sets. This is clear because $T^t$ is piecewise affine, so it maps any positive measure set into a finite union of positive measure sets. $\square$

## B.2. Proof of Theorem 3.1

Without loss of generality, we assume that $0 \in \mathcal{K}^0$ and thus $\mathsf{J}^t(x^\star) = \mathsf{J}^t(0) = 0$. Fix $i \in [\![1, n]\!]$, let

$$s_i^t := \arg\min_{y \in \mathcal{K}^t} \langle B_*^t x_i^t, y \rangle,$$

so that the update becomes

$$x_i^{t+1} = x_i^t + \gamma^t(s_i^t - x_i^t).$$

Expanding $\mathsf{J}^t(x_i^{t+1})$ exactly:

$$\mathsf{J}^t(x_i^{t+1}) = \frac{1}{2} \left\langle B_*^t \left( x_i^t + \gamma^t(s_i^t - x_i^t) \right), x_i^t + \gamma^t(s_i^t - x_i^t) \right\rangle$$

$$= \mathsf{J}^t(x_i^t) + \gamma^t \left\langle B_*^t x_i^t, s_i^t - x_i^t \right\rangle + \frac{(\gamma^t)^2}{2} \left\langle B_*^t(s_i^t - x_i^t), s_i^t - x_i^t \right\rangle.$$

By convexity of $\mathsf{J}^t$ and since $0 \in \mathcal{K}^t$, we have

$$\mathsf{J}^t(0) = 0 \le \mathsf{J}^t(y), \quad \forall y \in \mathcal{K}^t,$$

and in particular,

$$\langle B_*^t x_i^t, s_i^t - x_i^t \rangle \le -\mathsf{J}^t(x_i^t).$$

Moreover, since $s_i^t, x_i^t \in \mathcal{K}^t \subseteq \mathcal{K}^0$ and using the hypothesis on $B_*^t$, we have

$$\langle B_*^t(s_i^t - x_i^t), s_i^t - x_i^t \rangle \le \lambda_{\max}(B_*^0)\mathsf{d}(\mathcal{K}^0)^2.$$

Therefore,

$$\mathsf{J}^t(x_i^{t+1}) \le \mathsf{J}^t(x_i^t) - \gamma^t \mathsf{J}^t(x_i^t) + \frac{(\gamma^t)^2}{2}\lambda_{\max}(B_*^0)\mathsf{d}(\mathcal{K}^0)^2,$$

which simplifies to

$$\mathsf{J}^t(x_i^{t+1}) \le (1 - \gamma^t)\mathsf{J}^t(x_i^t) + \frac{(\gamma^t)^2}{2}\lambda_{\max}(B_*^0)\mathsf{d}(\mathcal{K}^0)^2.$$

Now, using the assumption $B_*^{t+1} \preccurlyeq B_*^t$ and $\mathcal{K}^{t+1} \subseteq \mathcal{K}^t$, we have

$$\mathsf{J}^{t+1}(x_i^{t+1}) \le \mathsf{J}^t(x_i^{t+1}),$$

so:

$$\mathsf{J}^{t+1}(x_i^{t+1}) \le (1 - \gamma^t)\mathsf{J}^t(x_i^t) + \frac{(\gamma^t)^2}{2}\lambda_{\max}(B_*^0)\mathsf{d}(\mathcal{K}^0)^2.$$

Let $C := \frac{1}{2}\lambda_{\max}(B_*^0)\mathsf{d}(\mathcal{K}^0)^2$, and recall $\gamma^t = \frac{2}{t+2}$. Then

$$\mathsf{J}^{t+1}(x_i^{t+1}) \le \left(1 - \frac{2}{t+2}\right)\mathsf{J}^t(x_i^t) + \frac{4}{(t+2)^2}C.$$

Define $a^t := (t+1)\mathsf{J}^t(x_i^t)$. Then

$$a^{t+1} = (t+2)\mathsf{J}^{t+1}(x_i^{t+1}) \le (t+2)\left(1 - \frac{2}{t+2}\right)\frac{a^t}{t+1} + 4C$$

$$= \frac{t}{t+1}a^t + 4C.$$

Inductively, with $a^0 = 0$, we obtain

$$a^t \le 4Ct \quad \Rightarrow \quad \mathsf{J}^t(x_i^t) \le \frac{4C}{t+1},$$

as claimed. $\qquad\square$

## B.3. Proof of Theorem 4.2

Note that the cell-assignment map $\sigma$ is well-defined since the cells $\mathscr{C}_i(v)$ have mutually disjoint interiors (Lemma 4.4). By definition of the cells $\mathscr{C}_i(v)$, none of the vertices $v_1, \ldots, v_\kappa$ move along the evolution of (SA$_\infty$). Generally, since $x_i^0 \in \mathscr{C}_{\sigma(i)}(v)$, the maximizer of $\langle Bx_i^0, y \rangle$ over $y \in \mathcal{K}$ is $v_{\sigma(i)}$. In other words,

$$\arg\max_{y \in \mathcal{K}}\langle Bx_i^0, y \rangle = v_{\sigma(i)},$$

and thus

$$x_i^1 = (1 - \gamma^0)x_i^0 + \gamma^0 v_{\sigma(i)}.$$

By convexity of $\mathscr{C}_{\sigma(i)}(v)$, and since both $x_i^0$ and $v_{\sigma(i)}$ lie in $\mathscr{C}_{\sigma(i)}(v)$, we conclude that $x_i^1 \in \mathscr{C}_{\sigma(i)}(v)$ (and in fact, remains in the interior of the cell). Repeating this argument inductively yields the stated formula. $\qquad\square$

## B.4. Proof of Lemma 4.4

We begin by showing the first point. Let $x_1, x_2 \in \mathscr{C}_i(v)$ and $\lambda \in [0,1]$. For all $y \in \mathcal{K}$, we have

$$\langle Bx_1, v_i \rangle \geq \langle Bx_1, y \rangle, \quad \langle Bx_2, v_i \rangle \geq \langle Bx_2, y \rangle.$$

Taking a convex combination,

$$\langle B(\lambda x_1 + (1-\lambda)x_2), v_i \rangle = \lambda \langle Bx_1, v_i \rangle + (1-\lambda)\langle Bx_2, v_i \rangle$$

and similarly for $\langle B(\lambda x_1 + (1-\lambda)x_2), y \rangle$. Thus,

$$\langle B(\lambda x_1 + (1-\lambda)x_2), v_i \rangle \geq \langle B(\lambda x_1 + (1-\lambda)x_2), y \rangle,$$

showing that $\lambda x_1 + (1-\lambda)x_2 \in \mathscr{C}_i(v)$. Hence $\mathscr{C}_i(v)$ is convex.

We now show the second point. Suppose $\operatorname{int}(\mathscr{C}_i(v) \cap \mathscr{C}_j(v)) \neq \varnothing$ for $i \neq j$. Then there exists $x \in \operatorname{int}(\mathscr{C}_i(v) \cap \mathscr{C}_j(v))$. In particular,

$$\langle Bx, v_i \rangle \geq \langle Bx, y \rangle \quad \text{and} \quad \langle Bx, v_j \rangle \geq \langle Bx, y \rangle \quad \forall y \in \mathcal{K}.$$

In particular, taking $y = v_j$ and $y = v_i$ respectively, we get

$$\langle Bx, v_i \rangle \geq \langle Bx, v_j \rangle, \quad \langle Bx, v_j \rangle \geq \langle Bx, v_i \rangle,$$

thus $\langle Bx, v_i \rangle = \langle Bx, v_j \rangle$. Now, if $B$ is full rank, the face

$$\mathsf{F}_{i \lozenge j} := \{x \in \mathcal{K} : \langle Bx, v_i - v_j \rangle = 0\}$$

is a hyperplane, hence has empty interior, contradicting the assumption that $x$ lies in the interior.

We conclude by showing the third point. Let $x \in \mathcal{K}$. Then, since $\{v_i\}_{i \in [\![1,\kappa]\!]}$ is a finite set, there exists $v_i$ such that

$$\langle Bx, v_i \rangle \geq \langle Bx, v_j \rangle \quad \forall j \in [\![1, \kappa]\!].$$

Thus $x \in \mathscr{C}_i(v)$, and so $\mathcal{K} \subseteq \bigcup_{i \in [\![1,\kappa]\!]} \mathscr{C}_i(v)$. The reverse inclusion is immediate since each $\mathscr{C}_i(v) \subseteq \mathcal{K}$ by definition. $\square$

## B.5. Proof of Proposition 4.5

For fixed $x \in \mathcal{K}$, the map $y \mapsto \langle Bx, y \rangle$ is linear. Since $\mathcal{K} = \operatorname{conv}\{v_1, \ldots, v_\kappa\}$, we have

$$\max_{y \in \mathcal{K}} \langle Bx, y \rangle = \max_{1 \leq j \leq \kappa} \langle Bx, v_j \rangle.$$

Therefore, by the definition of $\mathscr{C}_i(v)$, a point $x \in \mathcal{K}$ belongs to $\mathscr{C}_i(v)$ precisely when

$$\langle Bx, v_i \rangle \geq \langle Bx, v_j \rangle \qquad \text{for every } j = 1, \ldots, \kappa.$$

By assumption, $J(v_j) = c$ for every $j$, so we have $\|v_j\|_B^2 = 2c$. Using the polarization identity, we obtain, for each $j$,

$$2\langle Bx, v_j \rangle = \|x\|_B^2 + \|v_j\|_B^2 - \|x - v_j\|_B^2 = \|x\|_B^2 + 2c - \|x - v_j\|_B^2.$$

Subtracting the identities above for $v_i$ and $v_j$ gives

$$2\big(\langle Bx, v_i \rangle - \langle Bx, v_j \rangle\big) = \|x - v_j\|_B^2 - \|x - v_i\|_B^2.$$

Thus the inequalities

$$\langle Bx, v_i \rangle \geq \langle Bx, v_j \rangle, \qquad j = 1, \ldots, \kappa,$$

are equivalent to

$$\|x - v_i\|_B \leq \|x - v_j\|_B, \qquad j = 1, \ldots, \kappa.$$

It follows that

$$\mathscr{C}_i(v) = \{x \in \mathcal{K} : \|x - v_i\|_B \leq \|x - v_j\|_B \quad \forall j \in [\![1, \kappa]\!]\}.$$

The set on the right-hand side is exactly $\operatorname{Vor}_B(v_i) \cap \mathcal{K}$, proving the result. $\square$

### B.6. Proof of Theorem 4.9

We split the proof in three parts.

PART 1. EXISTENCE

Fix $i \in [\![1, n]\!]$ and consider

$$\begin{cases} x_\gamma^{t+1} = (1 - \gamma)x_\gamma^t + \gamma \arg\max_{y \in \mathcal{K}} \langle Bx_\gamma^t, y \rangle, \\ x_\gamma^0 = x_i^0 \in \mathrm{int}(\mathscr{C}_i(v)), \end{cases}$$

for $\gamma > 0$. Arguing as in the proof of Theorem 4.2, we gather that

$$x_\gamma^t \in \mathscr{C}_i(v) \quad \text{for all } t \geq 0.$$

Moreover, the evolution

$$x_\gamma^{t+1} - x_\gamma^t = \gamma(v_i - x_\gamma^t)$$

shows that the sequence $(x_\gamma^t)_{t \geq 0}$ moves along the straight line segment from $x_i^0$ toward $v_i$.

We sketch the argument to pass to the continuous limit as $\gamma \to 0$, which is mostly classical. We proceed as is always done for proving the convergence of the Euler method. Namely, we define two types of interpolants: the piecewise constant interpolant

$$\tilde{x}_\gamma(t) := x_\gamma^j \quad \text{for } t \in [j\gamma, (j+1)\gamma),$$

and the piecewise affine interpolant

$$\hat{x}_\gamma(t) := x_\gamma^j + \frac{t - j\gamma}{\gamma}(x_\gamma^{j+1} - x_\gamma^j) \quad \text{for } t \in [j\gamma, (j+1)\gamma).$$

Since the dynamics take place inside the compact set $\mathcal{K}$, there exists $M > 0$ such that

$$\|\tilde{x}_\gamma(t)\|, \|\hat{x}_\gamma(t)\| \leq M \quad \text{for all } t \geq 0.$$

Moreover, from the discrete evolution,

$$\left\| \frac{x_\gamma^{t+1} - x_\gamma^t}{\gamma} \right\| = \|v_i - x_\gamma^t\| \leq 2M,$$

so that both $\tilde{x}_\gamma$ and $\hat{x}_\gamma$ are uniformly Lipschitz continuous with Lipschitz constant $2M$ independent of $\gamma$. By the Arzelà–Ascoli theorem, the families $(\tilde{x}_\gamma)_{\gamma \geq 0}$ and $(\hat{x}_\gamma)_{\gamma \geq 0}$ are relatively compact in $C_{\mathrm{loc}}^0(\mathbb{R}_{\geq 0}; \mathbb{R}^d)$. Thus, up to extraction, both interpolants converge uniformly on compact intervals to a continuous curve $x(t)$. Moreover, for the affine interpolant $\hat{x}_\gamma$, we have

$$\dot{\hat{x}}_\gamma(t) = \frac{x_\gamma^{j+1} - x_\gamma^j}{\gamma} = v_i - x_\gamma^j,$$

which is uniformly bounded and converges uniformly to $v_i - x(t)$. Therefore, $\hat{x}_\gamma$ converges strongly in $W_{\mathrm{loc}}^{1,\infty}$ to $x$. Passing to the limit, the limiting curve $x$ satisfies the differential equation

$$\dot{x}(t) = v_i - x(t),$$

with initial condition $x(0) = x_i^0$. Thus, the motion follows the straight line joining $x_i^0$ to $v_i$, exponentially approaching $v_i$.

PART 2. UNIQUENESS

Note that the vector field $\mathsf{v} : \mathcal{K}^n \mapsto (\mathbb{R}^d)^n$, $\mathsf{v} = (\mathsf{v}^1, ..., \mathsf{v}^n)$, in (11), defined as

$$\mathsf{v}^i(x) := \arg\max_{y \in \{x_j\}_{j \in [\![1,n]\!]}} \langle Bx_i, y \rangle - x_i,$$

satisfies

$$\|\mathsf{v}\|_{L^\infty(\mathcal{K}^n;(\mathbb{R}^d)^n)} \leq 2\mathsf{d}(\mathcal{K}).$$

Define

$$\delta := \min_{j \in [\![1,n]\!]} \text{dist}\left(x_j^0, \partial \mathscr{C}_j(v) \cap \bigcup_{k \in \text{neigh}(\sigma(j))} \mathscr{C}_k(v)\right).$$

(Recall the definition of the neighbor vertices in (8).) Then, any solution to (11) satisfies

$$x_j(t) \in \mathsf{C}_{\sigma(j)}(v) \qquad \text{for } (t,j) \in \left[0, \frac{\delta}{2\mathsf{d}(\mathcal{K})}\right] \times [\![1,n]\!].$$

This means that, on the time interval $[0, \frac{\delta}{2\mathsf{d}(\mathcal{K})}]$, equation (11) reduces to

$$\dot{x}_j(t) = v_{\sigma(j)} - x_j(t). \tag{16}$$

By standard Cauchy–Lipschitz theory, the solution to (16) is unique and given by

$$x_j(t) = (1 - e^{-t})v_{\sigma(j)} + e^{-t}x_j(0), \qquad \text{for } t \in \left[0, \frac{\delta}{2\mathsf{d}(\mathcal{K})}\right]. \tag{17}$$

Now, define

$$\delta_j := \min_{t \in \mathbb{R}_{\geq 0}} \text{dist}\left(x_j(t), \partial \mathscr{C}_{\sigma(j)}(v) \cap \bigcup_{k \in \text{neigh}(\sigma(j))} \mathscr{C}_k(v)\right) > 0,$$

where $x_j(t)$ is the extension of the curve in (17) to the positive real line. This minimum is positive since (17) is a convex combination of $x_j(0)$ and $v_{\sigma(j)}$. We can now set $\delta^* = \min_j \delta_j$ and repeat the argument on the time interval

$$\left[\frac{\delta}{2\mathsf{d}(\mathcal{K})}, \frac{\delta}{2\mathsf{d}(\mathcal{K})} + \frac{\delta^*}{2\mathsf{d}(\mathcal{K})}\right],$$

with the solution again given by (17) on this interval. By the definition of $\delta^*$, we can iterate this argument *ad infinitum*, thus obtaining uniqueness for any time interval.

PART 3. CONTINUITY WITH RESPECT TO DATA

Consider $\tilde{x}_i^0 = x_i^0 + \Delta_i$ with $\|\Delta_i\| \leq \varepsilon$, and let $\tilde{v}_j$ denote the vertex of $\text{conv}\{\tilde{x}_i^0\}$ corresponding to $v_j$ for $\varepsilon$ small enough.

**Claim B.1.** *There exists $\varepsilon_* > 0$ such that for all $\varepsilon \in (0, \varepsilon_*)$ and $i \in [\![1,n]\!]$,*

$$\langle B\tilde{x}_i^0, \tilde{v}_{\sigma(i)}\rangle > \langle B\tilde{x}_i^0, y\rangle \quad \text{for all } y \in \left\{\tilde{x}_k^0\right\}_{k \in [\![1,n]\!]} \setminus \left\{\tilde{v}_{\sigma(i)}\right\}.$$

*Proof of Claim B.1.* Compute

$$\langle B\tilde{x}_i^0, \tilde{v}_{\sigma(i)}\rangle \geq \langle Bx_i^0, v_{\sigma(i)}\rangle - C_1(B)\left(\varepsilon + \varepsilon^2\right),$$

and, for any $y \neq \tilde{v}_{\sigma(i)}$,

$$\langle B\tilde{x}_i^0, y\rangle \leq \langle Bx_i^0, y\rangle + C_2(B)\left(\varepsilon + \varepsilon^2\right).$$

We find

$$\langle B\tilde{x}_i^0, \tilde{v}_{\sigma(i)}\rangle - \langle B\tilde{x}_i^0, y\rangle \geq \langle Bx_i^0, v_{\sigma(i)}\rangle - \langle Bx_i^0, y\rangle + O(\varepsilon).$$

Note that the first term of the right hand side is positive by uniqueness of the $\arg\max$ of the unperturbed problem, therefore, there exists some small enough $\varepsilon_* > 0$ for which the right hand side is positive. □

Therefore, for $\varepsilon$ small enough, the dynamics for $x_i$ simplify to

$$\dot{x}_i = \tilde{v}_{\sigma(i)} - x_i, \quad x_i(0) = \tilde{x}_i^0,$$

since $\tilde{x}_i^0$ lies in the region where $\tilde{v}_{\sigma(i)}$ is the unique maximizer. Viewing $\tilde{v}_{\sigma(i)}$ as a fixed parameter, we note that the equation is linear with constant coefficients. By standard continuity results for ODEs with respect to parameters and initial conditions, the resulting trajectory depends continuously on the perturbation, and thus remains close to the unperturbed one.

## B.7. Proof of Theorem 5.2

The proof is split in four steps.

### Step 1. Non-entry time

Consider the random variable

$$d_{ij}^t := \left\| x_j^t - v_i \right\|,$$

where $x_j^t$ corresponds to a realization of the process emanating from an initial particle $x_j^0$ lying in the cell $\mathscr{C}_i(v)$. In this first step, we look to show that

$$d_{ij}^t \geq (1-\gamma)^t d_{ij}^0 \qquad \text{with probability 1.} \tag{18}$$

From this we will deduce an estimate of the time up to which the particle $x_j^t$ cannot enter a ball centered at the vertex of the cell in which it lies.

We use the following purely geometric fact.

**Claim B.2.** *Fix $i \in [\![1, n]\!]$. Assume that for all $x, y \in \mathcal{K}$,*

$$\langle x - v_i, y - v_i \rangle > 0, \tag{19}$$

*Then, for every $x \in \mathcal{K}$,*

$$\arg\min_{y \in \mathcal{K}} \left\| (1-\gamma)x + \gamma y - v_i \right\| = v_i,$$

*Proof of Claim B.2.* Fix $x \in \mathcal{K}$ and set $a := x - v_i$. For any $y \in \mathcal{K}$ set $d := y - v_i$. Then

$$(1-\gamma)x + \gamma y - v_i = (1-\gamma)(x - v_i) + \gamma(y - v_i) = (1-\gamma)a + \gamma d.$$

Therefore,

$$\left\| (1-\gamma)a + \gamma d \right\|^2 - \left\| (1-\gamma)a \right\|^2 = 2\gamma(1-\gamma)\langle a, d \rangle + \gamma^2 \|d\|^2 \geq 0,$$

where the last inequality uses (19) and $\|d\|^2 \geq 0$. Hence, the minimum is attained at $d = 0$, i.e. $y = v_i$. $\qquad\square$

Note that

$$\left\| x_j^{t+1} - v_i \right\| = \left\| (1-\gamma)x_j^t + \gamma x_\ell^t - v_i \right\| \qquad \text{with probability } p_{j\to\ell}^t$$
$$\geq \left\| (1-\gamma)x_j^t + \gamma v_i - v_i \right\| \qquad \text{with probability 1,}$$

where the second inequality holds due to Claim B.2, and where

$$p_{j\to\ell}^t := \frac{e^{\beta\langle x_j^t, x_\ell^t \rangle}}{\displaystyle\sum_{\iota=1}^n e^{\beta\langle x_j^t, x_\iota^t \rangle}}.$$

We can repeat the above argument: with probability 1,

$$\left\| x_j^{t+1} - v_i \right\| \geq \left\| (1-\gamma)x_j^t + \gamma v_i - v_i \right\|$$
$$\geq \min_{y \in \mathcal{K}} \left\| (1-\gamma)((1-\gamma)x_j^{t-1} + \gamma y) + \gamma v_i - v_i \right\|$$
$$= (1-\gamma)\min_{v \in \mathcal{K}} \left\| (1-\gamma)x_j^{t-1} + \gamma v - v_i \right\|$$
$$= (1-\gamma)^2 \left\| x_j^{t-1} - v_i \right\|,$$

where in the last equality we use Claim B.2. Iterating yields (18).

Define

$$T_{1ji} := \left\lfloor \frac{1}{\log(1-\gamma)} \log\left( \frac{\tau}{\|x_j^0 - v_i\|} \right) \right\rfloor. \tag{20}$$

We recall that

$$\tau := \min_{i \in [\![1,\kappa]\!]} \frac{c_0}{2 \max_{j \neq i} \|v_i - v_j\|} \wedge \frac{\sqrt{2c_0}}{2}. \tag{21}$$

As a consequence of (18),

$$\mathbb{P}\left(x_j^t \notin B(v_i, \tau)\right) = 1 \qquad \text{for } t \in [\![0, T_{1ji}]\!].$$

Consider

$$T_1 := \min_{\substack{j \in [\![\kappa+1, n]\!] \\ i \in [\![1,\kappa]\!]}} T_{1ji}$$

$$= \left\lfloor \frac{1}{\log(1-\gamma)} \log\left(\frac{\tau}{\min_{j \in [\![\kappa+1,n]\!]} \|x_j^0 - v_{\sigma(j)}\|}\right) \right\rfloor;$$

the last equality follows from (13) and Proposition 4.5. One has

$$\mathbb{P}\left(x_j^t \notin B(v_i, \tau)\right) = 1 \qquad \text{for } t \in [\![0, T_1]\!], \text{ for all } j \neq i \text{ and } i \in [\![1, \kappa]\!].$$

**Step 2. Up to the non-entry time, vertices barely move**

Fix $i \in [\![1, \kappa]\!]$ and $t \in [\![0, T_1]\!]$, and define the random variable

$$M_i^t := \#\left\{s \in [\![1, t]\!]: x_i^s \neq x_i^{s-1}\right\}. \tag{22}$$

Recalling that the increments of the process (SA$_{\mathbb{P}}$) are bounded by $\gamma\mathsf{d}(\mathcal{K})$, and that $x_i^0 = v_i$, we have

$$\mathbb{P}\left(x_i^t \in B(v_i, \beta^{-\frac{1}{4}}) \,\middle|\, M_i^t < \frac{\beta^{-\frac{1}{4}}}{\gamma\mathsf{d}(\mathcal{K})}\right) = 1.$$

Hence

$$\mathbb{P}\left(x_i^t \in B(v_i, \beta^{-\frac{1}{4}})\right)$$

$$= \mathbb{P}\left(x_i^t \in B(v_i, \beta^{-\frac{1}{4}}) \,\middle|\, M_i^t < \frac{\beta^{-\frac{1}{4}}}{\gamma\mathsf{d}(\mathcal{K})}\right) \mathbb{P}\left(M_i^t < \frac{\beta^{-\frac{1}{4}}}{\gamma\mathsf{d}(\mathcal{K})}\right)$$

$$+ \mathbb{P}\left(x_i^t \in B(v_i, \beta^{-\frac{1}{4}}) \,\middle|\, M_i^t \geq \frac{\beta^{-\frac{1}{4}}}{\gamma\mathsf{d}(\mathcal{K})}\right) \mathbb{P}\left(M_i^t \geq \frac{\beta^{-\frac{1}{4}}}{\gamma\mathsf{d}(\mathcal{K})}\right)$$

$$\geq \mathbb{P}\left(M_i^t < \frac{\beta^{-\frac{1}{4}}}{\gamma\mathsf{d}(\mathcal{K})}\right).$$

To lower bound the last probability, we use

**Claim B.3.** *There exists $\beta_* > 0$ such that for $t \in [\![0, T_1]\!]$ and $i \in [\![1, n]\!]$ and $\beta \geq \beta_*$, conditioned on the event*

$$\left\{M_i^t < \frac{\beta^{-\frac{1}{4}}}{\gamma\mathsf{d}(\mathcal{K})}\right\},$$

*we have*

$$p_{i \to i}^t \geq 1 - ne^{-\beta\tau/4}.$$

*Proof of Claim B.3.* Since we condition on $M_i^t < \frac{\beta^{-\frac{1}{4}}}{\gamma\mathsf{d}(\mathcal{K})}$, we have $x_i^t \in B(v_i, \beta^{-\frac{1}{4}})$ with probability 1. Furthermore, since $t \in [\![0, T_1]\!]$, $x_j^t \notin B(v_i, \tau)$. Take an arbitrary $y \in B(v_i, \beta^{-\frac{1}{4}})$ and $\zeta \in B(v_i, \tau)^c \cap \mathcal{K}$.

We separately consider the cases $\zeta \in \mathscr{C}_i(v) \cap B(v_i, \tau)^c$ and $\zeta \notin \mathscr{C}_i(v)$. Let us start by $\zeta \in \mathscr{C}_i(v) \cap B(v_i, \tau)^c$. Then

$$\tau^2 \leq \|v_i - \zeta\|^2 = \|v_i\|^2 + \|\zeta\|^2 - 2\langle\zeta, v_i\rangle;$$

so
$$\tau^2 \leq \|v_i\|^2 + \|\zeta\|^2 - 2\langle \zeta, v_i \rangle.$$

Consequently
$$\langle v_i, \zeta \rangle \leq \frac{1}{2}\|v_i\|^2 + \frac{1}{2}\|\zeta\|^2 - \frac{\tau^2}{2} \leq \|v_i\|^2 - \frac{\tau^2}{2} \tag{23}$$

where the last inequality stems from $\zeta \in \mathscr{C}_i(v)$ and so $\|\zeta\| \leq \|v_i\|$. Indeed, $\zeta \in \mathscr{C}_i(v)$ implies $\langle \zeta, v_i \rangle \geq \langle \zeta, \zeta \rangle$, whilst $v_i \in \mathscr{C}_i(v)$ by (14) whereupon we gather $\langle \zeta, v_i \rangle \leq \langle v_i, v_i \rangle$.

We move to the other case: fix $\zeta \in \mathcal{K} \setminus \mathscr{C}_i(v)$ and consider the set
$$\mathcal{S} := \mathrm{conv}(\mathcal{K} \setminus \mathscr{C}_i(v)).$$

Because of the choice of $\tau$ in (21), we have $\mathcal{S} \cap B(v_i, \tau) = \varnothing$. We also have
$$\max_{\zeta \in \mathcal{K} \setminus \mathscr{C}_i(v)} \langle \zeta, v_i \rangle \leq \max_{\zeta \in \mathcal{S}} \langle \zeta, v_i \rangle.$$

Let $\zeta_* \in \mathcal{S}$ be the maximizer. Since $\mathcal{S}$ is a convex polytope,
$$\zeta_* = \sum_{j \neq i} m_j v_j + \sum_{k=1}^{P} m_k v_k, \qquad \text{with } \sum_{j \neq i} m_j + \sum_{k=1}^{P} m_k = 1$$

where $\{v_k\}_{k \in [P]}$ are the new vertices in $\mathcal{S}$ that lie on the boundary of $\mathscr{C}_i(v)$. Then, owing to (23) and (14) we have

$$\begin{aligned}
\langle \zeta_*, v_i \rangle &= \sum_{j \neq i} m_j \langle v_j, v_i \rangle + \sum_{k=1}^{P} m_k \langle v_k, v_i \rangle \\
&\leq (\|v_i\|^2 - c_0) \sum_{j \neq i} m_j + \left( \|v_i\|^2 - \frac{\tau^2}{2} \right) \sum_{k=1}^{P} m_k \\
&\leq \max \left\{ \|v_i\|^2 - c_0, \|v_i\|^2 - \frac{\tau^2}{2} \right\}.
\end{aligned}$$

By (21) we also have $(\|v_i\|^2 - c_0) \leq (\|v_i\|^2 - \tau^2/2)$, therefore
$$\langle \zeta, v_i \rangle \leq \|v_i\|^2 - \frac{\tau^2}{2} \qquad \text{for } \zeta \in \mathcal{K} \setminus B(v_i, \tau). \tag{24}$$

On the other hand, $y \in B(v_i, \beta^{-\frac{1}{4}})$, hence $y = v_i + \Delta$ for some $\|\Delta\| \leq \beta^{-\frac{1}{4}}$, and satisfies
$$\langle y, y \rangle \geq \|v_i\|^2 - 2\mathsf{d}(\mathcal{K})\beta^{-\frac{1}{4}} - \beta^{-\frac{1}{2}}. \tag{25}$$

Combining with (24), we have that
$$\langle y, \zeta \rangle = \langle v_i, \zeta \rangle + \langle \Delta, \zeta \rangle \leq \|v_i\|^2 - \frac{\tau^2}{2} + \beta^{-\frac{1}{4}}\mathsf{d}(\mathcal{K}). \tag{26}$$

Finally, gathering (25) and (26) we find
$$\langle y, y \rangle - \langle y, \zeta \rangle \geq \frac{\tau^2}{2} - 3\mathsf{d}(\mathcal{K})\beta^{-\frac{1}{4}} - \beta^{-\frac{1}{2}}. \tag{27}$$

Thus for $\beta$ large enough, using (27), we have

$$\begin{aligned}
p_{i \to i}^t &= \frac{e^{\beta\langle x_i^t, x_i^t \rangle}}{\displaystyle\sum_{j=1}^{n} e^{\beta\langle x_i^t, x_j^t \rangle}} = \frac{1}{1 + \displaystyle\sum_{j \neq i} e^{\beta(\langle x_i^t, x_j^t \rangle - \langle x_i^t, x_i^t \rangle)}} \geq \frac{1}{1 + (n-1)e^{\beta\tau/4}} \\
&\geq 1 - ne^{-\beta\tau/4}.
\end{aligned} \qquad \square$$

Set
$$p = p(\beta) := ne^{-\beta\tau/4}. \tag{28}$$

Using Claim B.3, we show that

$$\mathbb{P}\left(M_i^t < \frac{\beta^{-\frac{1}{4}}}{\gamma\mathsf{d}(\mathcal{K})}\right) \geq \sum_{m=0}^{\left\lfloor \frac{\beta^{-\frac{1}{4}}}{\gamma\mathsf{d}(\mathcal{K})} \right\rfloor} \mathbb{P}(\mathrm{Bin}(t+1, p) = m) \tag{29}$$

$$= \sum_{m=0}^{\left\lfloor \frac{\beta^{-\frac{1}{4}}}{\gamma\mathsf{d}(\mathcal{K})} \right\rfloor} \binom{t}{m} (1-p)^{t-m} p^m$$

$$= 1 - \sum_{\left\lfloor \frac{\beta^{-\frac{1}{4}}}{\gamma\mathsf{d}(\mathcal{K})} \right\rfloor}^{t} \binom{t}{m} (1-p)^{t-m} p^m,$$

where $\mathrm{Bin}(t, p)$ denotes the usual Binomial distribution. Inequality (29) follows from the following claim.

**Claim B.4.** *Let $\{b^s\}_{s=0}^t$ be independent Bernoulli random variables with parameters $\{p^s\}_{s=0}^t$. Assume that $p^s \geq p$ for all $s \in [\![0, t]\!]$, and define*

$$S = \sum_{s=0}^t b^s.$$

*Then, for all $\xi \in \mathbb{R}$, one has*

$$\mathbb{P}(S \leq \xi) \leq \mathbb{P}(\mathrm{Bin}(t+1, p) \leq \xi).$$

*Proof of Claim B.4.* Let $y^0, \ldots, y^t$ be i.i.d. Bernoulli random variables with parameter $p$, and define

$$B = \sum_{s=0}^t y^s \sim \mathrm{Bin}(t+1, p).$$

To prove the result, we introduce i.i.d. uniform random variables $u^0, \ldots, u^t$ on $[0, 1]$, for which it holds that

$$b^s = \mathbf{1}_{\{u^s \leq p^s\}}, \quad y^s = \mathbf{1}_{\{u^s \leq p\}}, \quad \forall s \in [\![0, t]\!].$$

Since $p^s \geq p$, we have $\{u^s \leq p\} \subseteq \{u^s \leq p^s\}$, hence $b^s \geq y^s$ a.s. in $s$ and therefore,

$$S = \sum_{s=0}^t b^s \geq \sum_{s=0}^t y^s = B \quad \text{a.s.}$$

We just need to check that $S \geq B$ a.s. implies the monotonicity for the cumulative density functions, that is,

$$\mathbb{P}(S \leq \xi) \leq \mathbb{P}(B \leq \xi).$$

Indeed, fix $\xi \in \mathbb{R}$ and define the events

$$C = \{\omega : S(\omega) \leq \xi\}, \quad D = \{\omega : B(\omega) \leq \xi\}.$$

Let $E = \{\omega : S(\omega) < B(\omega)\}$. Since $S \geq B$ a.s., we have $\mathbb{P}(E) = 0$. Now, if $S(\omega) \leq \xi$ and $S(\omega) \geq B(\omega)$, then $B(\omega) \leq \xi$ as well, and thus $C \setminus D \subseteq E$. This implies that $\mathbb{P}(C \setminus D) = 0$ and we obtain

$$\mathbb{P}(S \leq \xi) = \mathbb{P}(C) = \mathbb{P}(C \cap D) + \mathbb{P}(C \setminus D) \leq \mathbb{P}(D) = \mathbb{P}(B \leq \xi),$$

as desired. □

Since

$$\sum_{m=\left\lceil \frac{\beta^{-\frac{1}{4}}}{\gamma d(\mathcal{K})} \right\rceil}^{t} \binom{t}{m} (1-p)^{t-m} p^m \leq p^{\left\lceil \frac{\beta^{-\frac{1}{4}}}{\gamma d(\mathcal{K})} \right\rceil} 2^t$$

we conclude that

$$\mathbb{P}\left( x_i^t \in B(v_i, \beta^{-\frac{1}{4}}) \right) \geq 1 - p^{\left\lceil \frac{\beta^{-\frac{1}{4}}}{\gamma d(\mathcal{K})} \right\rceil} 2^t \qquad \text{for } (t, i) \in [\![0, T_1]\!] \times [\![1, \kappa]\!]. \tag{30}$$

Due to the form of $p$, this already yields one part of the statement, should $\beta$ be large enough.

**Step 3. Probability that an interior point drifts away from its origin**

We proceed similarly in bounding

$$\mathbb{P}\left( x_j^t \in \mathcal{I}_{\sigma(j)}(\beta^{-\frac{1}{8}}) \,\middle|\, x_j^0 \in \mathcal{I}_{\sigma(j)}(\beta^{-\frac{1}{8}}) \ominus \beta^{-\frac{1}{4}} B_1, \, x_i^t \in B(v_i, \beta^{-\frac{1}{4}}), \text{ for all } i \in [\![1, \kappa]\!] \right)$$

for $t \in [\![0, T_1]\!]$ and $j \in [\![\kappa+1, n]\!]$. We prove and use a couple of facts. The first one is purely geometric.

**Claim B.5.** *Fix $i \in [\![1, \kappa]\!]$. For any $\eta > 0$, $x \in \mathcal{I}_i(\eta)$ and $z \in \mathcal{K} \setminus B(v_i, \tau)$, we have*

$$\langle x, z \rangle \leq \langle x, v_i \rangle - \frac{\eta \tau}{d(\mathcal{K})}.$$

*Proof of Claim B.5.* Any point $z \in \mathcal{K}$ can be written as

$$z = \sum_{j=1}^{\kappa} \lambda_j v_j, \quad \text{where } \lambda_j \geq 0, \quad \sum_{j=1}^{\kappa} \lambda_j = 1,$$

thus

$$\langle x, z \rangle = \sum_{j=1}^{\kappa} \lambda_j \langle x, v_j \rangle = \langle x, v_i \rangle - \sum_{j \neq i} \lambda_j \langle x, v_i - v_j \rangle.$$

Since $x \in \mathcal{I}_i(\eta)$, we have

$$\langle x, z \rangle \leq \langle x, v_i \rangle - \eta \sum_{j \neq i} \lambda_j = \langle x, v_i \rangle - \eta(1 - \lambda_i).$$

Then

$$\|z - v_i\| = \left\| \sum_{j \neq i} \lambda_j (v_j - v_i) \right\| \leq \sum_{j \neq i} \lambda_j \|v_j - v_i\| \leq d(\mathcal{K})(1 - \lambda_i).$$

As $\|z - v_i\| \geq \tau$, we have $1 - \lambda_i \geq \frac{\tau}{d(\mathcal{K})}$, and substituting into the earlier bound yields the claim. $\qquad \square$

We crucially need

**Claim B.6.** *There exists $\beta_* > 0$ such that for $t \in [\![0, T_1]\!]$, conditioned on*

$$\left\{ x_i^t \in B(v_i, \beta^{-\frac{1}{4}}) \text{ for all } i \in [\![1, \kappa]\!] \right\},$$

*we have that*

$$p_{j \to \sigma(j)}^t \geq 1 - n e^{-\beta^{\frac{7}{8}} \tau / 2}.$$

*Proof of Claim B.6.* Fix $x_j \in \mathcal{I}_{\sigma(j)}(\beta^{-\frac{1}{8}})$. For $y \in B(v_{\sigma(j)}, \beta^{-\frac{1}{4}})$, as $y = v_{\sigma(j)} + \Delta$ with $\|\Delta\| \leq \beta^{-\frac{1}{4}}$, we have

$$\langle x_j, y \rangle \geq \langle x_j, v_{\sigma(j)} \rangle - \beta^{-\frac{1}{4}} \|x_j\| \geq \langle x_j, v_{\sigma(j)} \rangle - \beta^{-\frac{1}{4}} \mathsf{d}(\mathcal{K}). \tag{31}$$

On the other hand, by virtue of Claim B.5, we have

$$\langle x_j, z \rangle \leq \langle x_j, v_{\sigma(j)} \rangle - \tau \beta^{-\frac{1}{8}} \tag{32}$$

for all $z \in \mathcal{K} \setminus B(v_{\sigma(j)}, \tau)$. Using (31) and (32) we obtain

$$\langle x_j, y \rangle - \langle x_j, z \rangle \geq -\tau \beta^{-\frac{1}{8}} + \beta^{-\frac{1}{4}} \mathsf{d}(\mathcal{K}),$$

which implies that for $\beta$ large enough,

$$\langle x_j, y \rangle - \langle x_j, z \rangle \geq -\beta^{-\frac{1}{8}} \frac{\tau}{2}.$$

This implies

$$p^t_{j \to \sigma(j)} \geq 1 - n e^{-\beta^{\frac{7}{8}} \tau/2},$$

as desired. $\qquad\qquad\square$

Set

$$\widetilde{p} := n e^{-\beta^{\frac{7}{8}} \tau/2}.$$

Following the same arguments as in the previous step, we can deduce

$$\mathbb{P}\Bigg( x^t_j \in \mathcal{I}_{\sigma(j)}(\beta^{-\frac{1}{8}}) \,\Bigg|\, x^0_j \in \mathcal{I}_{\sigma(j)}(\beta^{-\frac{1}{8}}) \ominus \beta^{-\frac{1}{4}} B_1,$$

$$x^t_i \in B(v_i, \beta^{-\frac{1}{4}}), \text{ for all } i \in [\![1, \kappa]\!] \Bigg)$$

$$\geq 1 - \widetilde{p}^{\left\lfloor \frac{\beta^{-\frac{1}{4}}}{\gamma \mathsf{d}(\mathcal{K})} \right\rfloor} 2^t,$$

and using (30) we also obtain

$$\mathbb{P}\left( x^t_j \in \mathcal{I}_{\sigma(j)}(\beta^{-\frac{1}{8}}) \,\Bigg|\, x^0_j \in \mathcal{I}_{\sigma(j)}(\beta^{-\frac{1}{8}}) \ominus \beta^{-\frac{1}{4}} B_1 \right)$$

$$\geq \mathbb{P}\Bigg( x^t_j \in \mathcal{I}_{\sigma(j)}(\beta^{-\frac{1}{8}}) \,\Bigg|\, x^0_j \in \mathcal{I}_{\sigma(j)}(\beta^{-\frac{1}{8}}) \ominus \beta^{-\frac{1}{4}} B_1,$$

$$x^t_i \in B(v_i, \beta^{-\frac{1}{4}}) \text{ for all } i \in [\![1, \kappa]\!] \Bigg)$$

$$\mathbb{P}\left( x^t_i \in B(v_i, \beta^{-\frac{1}{4}}) \text{ for all } i \in [\![1, \kappa]\!] \right)$$

$$\geq \left( 1 - \widetilde{p}^{\left\lfloor \frac{\beta^{-\frac{1}{4}}}{\gamma \mathsf{d}(\mathcal{K})} \right\rfloor} 2^t \right) \left( 1 - \kappa p^{\left\lfloor \frac{\beta^{-\frac{1}{4}}}{\gamma \mathsf{d}(\mathcal{K})} \right\rfloor} 2^t \right). \tag{33}$$

**Step 4. Reaching a ball centered at the vertex**

Here we look to estimate

$$\mathbb{P}\left( x^{T_1}_j \in B\left(v_{\sigma(j)}, C\tau\right) \setminus B\left(v_{\sigma(j)}, \tau\right) \,\Bigg|\, \Omega \right)$$

for some numerical $C > 1$ to be determined later, where

$$\Omega := \left\{ x_i^t \in B\left(v_i, \beta^{-\frac{1}{4}}\right) \quad \forall i \in [\![1, \kappa]\!], \right.$$

$$\left. x_j^t \in \mathscr{I}_{\sigma(j)}(\beta^{-\frac{1}{8}}) \quad \forall j \in [\![\kappa + 1, n]\!], \ \forall t \in [\![0, T_1]\!] \right\}.$$

To simplify notation in this step, all expectations, variances, and probabilities are conditioned on $\Omega$ without explicit mention. However, we emphasize that the probabilities can be estimated directly using (30) and (33).

STEP 4.1. EXPECTATION OF THE CONTRACTION

By Claim B.2, Claim B.3, and Claim B.6,

$$d_{ij}^{t+1} \geq (1 - \gamma)d_{ij}^t \quad \text{with probability 1,} \tag{34}$$

while

$$d_{ij}^{t+1} \leq (1 - \gamma)(d_{ij}^t - \beta^{-\frac{1}{4}}) \quad \text{with probability at least } 1 - p, \tag{35}$$

and

$$d_{ij}^{t+1} \leq d_{ij}^t + \gamma \, \mathsf{d}(\mathcal{K}) \quad \text{with probability at most } p, \tag{36}$$

where $p = p(\beta)$ is as in (28).

Indeed, (36) follows from the bound $\|x_j^t\| \leq \mathsf{d}(\mathcal{K})$.

To justify (35), fix $x \in \mathcal{K} \setminus B(v_i, \beta^{-\frac{1}{4}})$. Then

$$\min_{v \in B(v_i, \beta^{-\frac{1}{4}})} \|(1 - \gamma)x + \gamma v - v_i\| = \min_{v \in B(v_i, \beta^{-\frac{1}{4}})} \|(1 - \gamma)(x - v_i) + \gamma(v - v_i)\|$$

$$= \min_{w \in B(0, \beta^{-\frac{1}{4}})} \|(1 - \gamma)z + \gamma w\|,$$

where $z := x - v_i$. Since $z \notin B(0, \beta^{-\frac{1}{4}})$, the minimum is attained at $w = \beta^{-\frac{1}{4}} \frac{z}{\|z\|}$, yielding

$$\left\|(1 - \gamma)z + \gamma \beta^{-\frac{1}{4}} \frac{z}{\|z\|}\right\| = \frac{1}{\|z\|}\|((1 - \gamma)\|z\| + \gamma \beta^{-\frac{1}{4}})z\| = (1 - \gamma)\|z\| + \gamma \beta^{-\frac{1}{4}}.$$

Reversing the change of variables, we obtain

$$d_{ij}^{t+1} \leq (1 - \gamma)d_{ij}^t + \gamma \beta^{-\frac{1}{4}} \quad \text{with probability at least } 1 - p.$$

Combining (34), (35), and (36), we find that

$$(1 - \gamma)^t d_{ij}^0 \leq \mathbb{E}\left[d_{ij}^t\right]$$

$$\leq ((1 - p)(1 - \gamma) + p)^t d_{ij}^0$$

$$+ \sum_{s=0}^{t} ((1 - p)(1 - \gamma) + p)^s \left(p\gamma \mathsf{d}(\mathcal{K}) - (1 - p)(1 - \gamma)\beta^{-\frac{1}{4}}\right). \tag{37}$$

STEP 4.2. BOUNDING THE VARIANCE

We want to bound the variance of $d_{ij}^{t+1}$. To do so, we express the variance of $d_{ij}^{t+1}$ in terms of the variance of $d_{ij}^t$ using the law of total variance:

$$\mathrm{Var}\left(d_{ij}^{t+1}\right) = \mathbb{E}\left[\mathrm{Var}\left(d_{ij}^{t+1} \mid d_{ij}^t\right)\right] + \mathrm{Var}\left(\mathbb{E}\left[d_{ij}^{t+1} \mid d_{ij}^t\right]\right).$$

We first address the first term by noting that

$$\mathrm{Var}\left(d_{ij}^{t+1} \mid d_{ij}^t\right) = \mathbb{E}\left[\left(d_{ij}^{t+1}\right)^2 \mid d_{ij}^t\right] - \mathbb{E}\left[d_{ij}^{t+1} \mid d_{ij}^t\right]^2.$$

Letting $p = p(\beta)$ be as in (28), we have

$$\mathbb{E}\left[\left(d_{ij}^{t+1}\right)^2 \mid d_{ij}^t\right] \leq \left[(1-p)(1-\gamma)^2 + p\right]\left(d_{ij}^t\right)^2$$
$$+ \left[2(1-\gamma)\gamma(1-p) + 2p\gamma\mathsf{d}(\mathcal{K})\right]d_{ij}^t$$
$$+ \left[(1-p)\gamma^2\beta^{-\frac{1}{2}} + p\gamma^2\mathsf{d}(\mathcal{K})^2\right].$$

On the other hand, by the same arguments as in Claim B.2, we have

$$\mathbb{E}\left[d_{ij}^{t+1} \mid d_{ij}^t\right] \geq (1-\gamma)d_{ij}^t.$$

Hence,

$$\mathrm{Var}\left(d_{ij}^{t+1} \mid d_{ij}^t\right) \leq p\gamma(2-\gamma)\left(d_{ij}^t\right)^2$$
$$+ 2\gamma\left[(1-\gamma)\beta^{-\frac{1}{4}}(1-p) + p\mathsf{d}(\mathcal{K})\right]d_{ij}^t$$
$$+ (1-p)\gamma^2\beta^{-\frac{1}{2}} + p\gamma^2\mathsf{d}(\mathcal{K})^2. \tag{38}$$

Therefore,

$$\mathbb{E}\left[\mathrm{Var}\left(d_{ij}^{t+1} \mid d_{ij}^t\right)\right] \leq p\gamma(2-\gamma)\mathbb{E}\left[\left(d_{ij}^t\right)^2\right]$$
$$+ 2\gamma\left[(1-\gamma)\beta^{-\frac{1}{4}}(1-p) + p\mathsf{d}(\mathcal{K})\right]\mathbb{E}\left[d_{ij}^t\right]$$
$$+ (1-p)\gamma^2\beta^{-\frac{1}{2}} + p\gamma^2\mathsf{d}(\mathcal{K})^2. \tag{39}$$

We now address the second term:

$$\mathrm{Var}\left(\underbrace{\mathbb{E}\left[d_{ij}^{t+1} \mid d_{ij}^t\right]}_{:=\varphi}\right) = \mathbb{E}\left[\varphi^2\right] - \mathbb{E}\left[\varphi\right]^2.$$

Proceeding similarly, we find

$$\mathbb{E}\left[\varphi^2\right]$$
$$\leq \left[p^2 + (1-p)^2(1-\gamma)^2 + 2p(1-p)(1-\gamma)\right]\mathbb{E}\left[\left(d_{ij}^t\right)^2\right]$$
$$+ \left[2p^2\gamma\mathsf{d}(\mathcal{K}) + 2(1-p)^2(1-\gamma)\gamma\beta^{-\frac{1}{4}} + 2p(1-p)(\gamma\beta^{-\frac{1}{4}} + \gamma\mathsf{d}(\mathcal{K}))\right]\mathbb{E}\left[d_{ij}^t\right]$$
$$+ \left[p^2\gamma\mathsf{d}(\mathcal{K})^2 + 2p(1-p)\gamma^2\beta^{-\frac{1}{4}}\mathsf{d}(\mathcal{K}) + (1-p)^2\gamma^2\beta^{-\frac{1}{2}}\right],$$

while

$$\mathbb{E}\left[\varphi\right]^2 \geq (1-\gamma)^2\mathbb{E}\left[d_{ij}^t\right]^2.$$

Thus,

$$\mathrm{Var}\left[\varphi\right]$$
$$\leq \left[p^2 + (1-p)^2(1-\gamma)^2 + 2p(1-p)(1-\gamma)\right]\mathbb{E}\left[\left(d_{ij}^t\right)^2\right]$$
$$- (1-\gamma)^2\mathbb{E}\left[d_{ij}^t\right]^2$$
$$+ \left[2p^2\gamma\,\mathsf{d}(\mathcal{K}) + 2(1-p)^2(1-\gamma)\gamma\beta^{-\frac{1}{4}} + 2p(1-p)(\gamma\beta^{-\frac{1}{4}} + \gamma\,\mathsf{d}(\mathcal{K}))\right]\mathbb{E}\left[d_{ij}^t\right]$$
$$+ \left[p^2\gamma\,\mathsf{d}(\mathcal{K})^2 + 2p(1-p)\gamma^2\beta^{-\frac{1}{4}}\mathsf{d}(\mathcal{K}) + (1-p)^2\gamma^2\beta^{-\frac{1}{2}}\right]. \tag{40}$$

Combining (39) and (40), and using the inequality

$$a\mathbb{E}\left[x^2\right] - b\mathbb{E}[x]^2 \leq b\,\mathrm{Var}[x] + (a-b)\mathbb{E}\left[x^2\right],$$

along with the bounds $(1 - \gamma) \leq 1$, $(1 - p) \leq 1$, and $\mathbb{E}\left[d_{ij}^t\right] \leq \mathsf{d}(\mathcal{K})$, we obtain

$$
\begin{aligned}
&\mathrm{Var}\left[d_{ij}^{t+1}\right] \\
&\leq (1 - \gamma)^2 \, \mathrm{Var}\left[d_{ij}^t\right] \\
&\quad + p\left[p + (p - 2)(1 - \gamma)^2 + 2(1 - p)(1 - \gamma) + \gamma(2 - \gamma)\right] \mathbb{E}\left[d_{ij}^t\right]^2 \\
&\quad + \beta^{-\frac{1}{4}}\left[3\gamma\,\mathsf{d}(\mathcal{K}) + \gamma^2 + 2\gamma\beta^{-\frac{1}{4}} + 2\gamma^2\beta^{-\frac{1}{4}}\right] \\
&\quad + p\Big[\mathsf{d}(\mathcal{K})^2 + 2\gamma\,\mathsf{d}(\mathcal{K})^2 + 2p\gamma\,\mathsf{d}(\mathcal{K})^2 + 2\gamma\beta^{-\frac{1}{4}}\mathsf{d}(\mathcal{K}) + 2\gamma\,\mathsf{d}(\mathcal{K})^2 \\
&\qquad + p\gamma\,\mathsf{d}(\mathcal{K}) + 2\gamma^2\beta^{-\frac{1}{4}}\mathsf{d}(\mathcal{K}) + \gamma^2\mathsf{d}(\mathcal{K})^2\Big] \\
&\leq (1 - \gamma)^2 \, \mathrm{Var}\left[d_{ij}^t\right] + C_0 p + C_1 \beta^{-\frac{1}{4}},
\end{aligned}
$$

where $C_0$ and $C_1$ depend on $h, \mathsf{d}(\mathcal{K}), \beta$, and $p$ as in the inequality above. Furthermore, $C_0$ and $C_1$ remain uniformly bounded as $\beta \to +\infty$, and $\gamma \to 0$. All in all,

$$
\mathrm{Var}\left[d_{ij}^{t+1}\right] \leq (1 - \gamma)^{2t}\mathrm{Var}\left[d_{ij}^1\right] + \sum_{s=0}^{t}(1 - \gamma)^{2s}\left(C_0 p + C_1\beta^{-\frac{1}{4}}\right).
$$

STEP 4.3. CONCENTRATION

By Chebyshev's inequality, we have

$$
\begin{aligned}
&\mathbb{P}\left(\left|d_{ij}^t - \mathbb{E}\left[d_{ij}^t\right]\right| \geq \varepsilon\right) \\
&\leq \frac{\mathrm{Var}\left[d_{ij}^t\right]}{\varepsilon^2} \\
&\leq \frac{(1 - \gamma)^{2t}\mathrm{Var}\left[d_{ij}^1\right] + \displaystyle\sum_{s=0}^{t}(1 - \gamma)^{2s}\left(C_0 p + C_1\beta^{-\frac{1}{4}}\right)}{\varepsilon^2} \\
&\leq \frac{\mathrm{Var}\left[d_{ij}^1\right] + t\left(C_0 p + C_1\beta^{-\frac{1}{4}}\right)}{\varepsilon^2},
\end{aligned}
\tag{41}
$$

and

$$
\begin{aligned}
&\mathrm{Var}\left[d_{ij}^1\right] \\
&\leq p\gamma(2 - \gamma)\mathsf{d}(\mathcal{K})^2 + \left(2\gamma\,\mathsf{d}(\mathcal{K}) + \gamma^2\beta^{-\frac{1}{4}}\right)\beta^{-\frac{1}{4}} + \left(2\gamma\,\mathsf{d}(\mathcal{K})^2 + \gamma^2\mathsf{d}(\mathcal{K})^2\right)p,
\end{aligned}
$$

where the inequality follows similarly to (38), replacing[4] $d_{ij}^t$ by $d_{ij}^0$. From (37), we obtain

$$
(1 - \gamma)^t d_{ij}^0 \leq \mathbb{E}\left[d_{ij}^t\right] \leq (1 + \gamma p)^t d_{ij}^0 + t\left[p\gamma\,\mathsf{d}(\mathcal{K}) + (1 - p)(1 - \gamma)\beta^{-\frac{1}{4}}\right].
$$

Evaluating at

$$
t = T_1 = \left\lceil \frac{1}{\log(1 - \gamma)} \log\left(\frac{\tau}{\min_{\ell \in [\![\kappa+1, n]\!]} d_{\ell\sigma(\ell)}^0}\right)\right\rceil,
$$

---

[4]Note that in the law of total variance, the second term is absent since $d_{ij}^0$ is fixed. Therefore, $\mathbb{E}\left[d_{ij}^1 \mid d_{ij}^0\right]$ is not a random variable.

we obtain, for $\beta$ large enough,

$$\frac{\tau \, d_{ij}^0}{\min_{\ell \in [\![\kappa+1,n]\!]} d_{\ell\sigma(\ell)}^0} \leq \mathbb{E}\left[d_{ij}^t\right] \leq \left(\frac{\tau}{\min_{\ell \in [\![\kappa+1,n]\!]} d_{\ell\sigma(\ell)}^0}\right)^{1+O(p)} d_{ij}^0$$
$$+ T_1\left(p\gamma \, \mathsf{d}(\mathcal{K}) + \beta^{-\frac{1}{4}}\right)$$
$$\leq (1+O(p))\frac{\tau \, d_{ij}^0}{\min_{\ell \in [\![\kappa+1,n]\!]} d_{\ell\sigma(\ell)}^0}$$
$$+ T_1\left(p\gamma \, \mathsf{d}(\mathcal{K}) + \beta^{-\frac{1}{4}}\right).$$

Since (13) holds, we know by (4.5) that the cells are Voronoi cells. Therefore, the minimum in (20) is attained for some pair $(\ell, \sigma(\ell))$. Fix $\ell \in [\![\kappa+1,n]\!]$ to be the minimizer of (20). Let

$$C := \max_{i \in [\![1,n]\!]} \max_{j \in [\![\kappa+1,n]\!]} \frac{d_{ij}^0}{\min_{\ell \in [\![\kappa+1,n]\!]} d_{\ell\sigma(\ell)}^0}.$$

STEP 4.4. CONCLUSION

Gathering (30), (33), and (41), we obtain

$$d_{ij}^{T_1} \in \left[\frac{\tau \, d_{ij}^0}{\min_{\ell \in [\![\kappa+1,n]\!]} d_{\ell\sigma(\ell)}^0}, \frac{\tau \, d_{ij}^0}{\min_{\ell \in [\![\kappa+1,n]\!]} d_{\ell\sigma(\ell)}^0} + O\left(T_1 p\right) + O(T_1 \beta^{-\frac{1}{4}}) + \varepsilon\right]$$

for all $j \in [\![\kappa+1,n]\!]$, with probability at least

$$\mathbb{P}\left(\bigcap_{j \in [\![\kappa+1,n]\!]} \left\{x_j^{T_1} \in B\left(v_{\sigma(j)}, C\tau\right) \setminus B\left(v_{\sigma(j)}, \tau\right)\right\}\right)$$
$$\geq \mathbb{P}\left(\bigcap_{j \in [\![\kappa+1,n]\!]} \left\{x_j^{T_1} \in B\left(v_{\sigma(j)}, C\tau\right) \setminus B\left(v_{\sigma(j)}, \tau\right)\right\} \Big| \Omega\right) \mathbb{P}(\Omega)$$
$$= \mathbb{P}\left(\bigcap_{j \in [\![\kappa+1,n]\!]} \left\{x_j^{T_1} \in B\left(v_{\sigma(j)}, C\tau\right) \setminus B\left(v_{\sigma(j)}, \tau\right)\right\} \cap \Omega.\right)$$

Note that the event in the last expression is precisely the event in the statement of the theorem. Now,

$$\mathbb{P}\left(\bigcap_{j \in [\![\kappa+1,n]\!]} \left\{x_j^{T_1} \in B\left(v_{\sigma(j)}, C\tau\right) \setminus B\left(v_{\sigma(j)}, \tau\right)\right\} \Big| \Omega\right) \mathbb{P}(\Omega)$$
$$= \left(1 - \mathbb{P}\left(\bigcup_{j \in [\![\kappa+1,n]\!]} \left\{x_j^{T_1} \in B\left(v_{\sigma(j)}, C\tau\right) \setminus B\left(v_{\sigma(j)}, \tau\right)\right\} \Big| \Omega\right)\right) \mathbb{P}(\Omega)$$
$$\geq \left(1 - (n-\kappa) \max_{j \in [\![\kappa+1,n]\!]} \mathbb{P}\left(x_j^{T_1} \in B\left(v_{\sigma(j)}, C\tau\right) \setminus B\left(v_{\sigma(j)}, \tau\right) \Big| \Omega\right)\right) \mathbb{P}(\Omega).$$

Using (41) with $t = T_1$, we obtain a lower bound for

$$\mathbb{P}\left(x_j^{T_1} \in B\left(v_{\sigma(j)}, C\tau\right) \setminus B\left(v_{\sigma(j)}, \tau\right) \Big| \Omega\right).$$

At the same time, we can obtain a lower bound for $\mathbb{P}(\Omega)$ by (33) and (30), ending up with

$$
\mathbb{P}\left(\bigcap_{j\in[\![\kappa+1,n]\!]}\left\{x_j^{T_1}\in B\left(v_{\sigma(j)},C\tau\right)\setminus B\left(v_{\sigma(j)},\tau\right)\right\}\bigg|\,\Omega\right)\mathbb{P}(\Omega)
$$

$$
\geq\left(1-(n-\kappa)\frac{O\left(T_1p\right)+O(T_1\beta^{-\frac14})}{\varepsilon^2}\right)\mathbb{P}(\Omega)
$$

$$
\geq\left(1-(n-\kappa)\frac{O\left(T_1p\right)+O(T_1\beta^{-\frac14})}{\varepsilon^2}\right)\left(1-\widetilde{p}^{\frac{\beta^{-\frac14}}{\gamma\,\mathsf{d}(\mathcal{K})}}2^{T_1}\right)^{n-\kappa}\left(1-\kappa p^{\frac{\beta^{-\frac14}}{\gamma\,\mathsf{d}(\mathcal{K})}}2^{T_1}\right). \tag{42}
$$

For $\beta$ large enough, expanding $\mathbb{P}(\Omega)$, we obtain

$$
1-\mathbb{P}(\Omega)\leq 1-\left(1-\widetilde{p}^{\frac{\beta^{-\frac14}}{\gamma\,\mathsf{d}(\mathcal{K})}}2^{T_1}\right)^{n-\kappa}\left(1-\kappa p^{\frac{\beta^{-\frac14}}{\gamma\,\mathsf{d}(\mathcal{K})}}2^{T_1}\right)
$$

$$
\lesssim 1-\left(1-(n-\kappa)\widetilde{p}^{\frac{\beta^{-\frac14}}{\gamma\,\mathsf{d}(\mathcal{K})}}2^{T_1}\right)\left(1-\kappa p^{\frac{\beta^{-\frac14}}{\gamma\,\mathsf{d}(\mathcal{K})}}2^{T_1}\right)
$$

$$
\lesssim (n-\kappa)\exp\left(-\beta^{\frac78}\frac{\tau}{2}\frac{\beta^{-\frac14}}{\gamma\,\mathsf{d}(\mathcal{K})}+\log(2)T_1\right)
$$

$$
+\kappa\exp\left(-\beta\frac{\tau}{2}\frac{\beta^{-\frac14}}{\gamma\,\mathsf{d}(\mathcal{K})}+\log(2)T_1\right)
$$

$$
\lesssim n\wedge(n-\kappa)\exp\left(-\beta^{\frac58}\frac{\tau}{2\gamma\,\mathsf{d}(\mathcal{K})}+O(T_1)\right)
$$

$$
=\exp\left(-\beta^{\frac58}\frac{\tau}{2\gamma\,\mathsf{d}(\mathcal{K})}+O(T_1)+2\log(n\wedge(n-\kappa))-O(1)\right).
$$

All in all,

$$
\mathbb{P}\left(\bigcap_{j\in[\![\kappa+1,n]\!]}\left\{x_j^{T_1}\in B\left(v_{\sigma(j)},C\tau\right)\setminus B\left(v_{\sigma(j)},\tau\right)\right\}\right)
$$

$$
\geq 1-(n-\kappa)\frac{O\left(T_1p\right)+O(T_1\beta^{-\frac14})}{\varepsilon^2}
$$

$$
-\exp\left(-\beta^{\frac58}\frac{\tau}{2\gamma\,\mathsf{d}(\mathcal{K})}+O(T_1)+2\log(n\wedge(n-\kappa))-O(1)\right).
$$

Choosing $\varepsilon=\beta^{-\frac{1}{32}}$, we can take $\beta$ large enough to obtain the bound in the statement of the theorem. This concludes the proof. $\qquad\square$

*Remark* B.7. We can improve (30) by applying a Chernoff inequality for the binomial distribution. Fix

$$
\alpha:=\left\lfloor\frac{\beta^{-\frac14}}{\gamma\,\mathsf{d}(\mathcal{K})}+1\right\rfloor
$$

and $\lambda>0$. Then,

$$
\mathbb{P}\left(\mathrm{Bin}(t,p)\geq\alpha\right)=\mathbb{P}\left(e^{\lambda\,\mathrm{Bin}(t,p)}\geq e^{\lambda\alpha}\right).
$$

By Markov's inequality, we obtain

$$
\mathbb{P}\left(\mathrm{Bin}(t,p)\geq\alpha\right)\leq e^{-\lambda\alpha}\,\mathbb{E}\left[e^{\lambda\,\mathrm{Bin}(t,p)}\right].
$$

At the same time, since the binomial is the sum of $t$ independent Bernoulli random variables, denoted $X_s \sim \mathrm{Bern}(p)$ for $s = 1, \ldots, t$, we compute:

$$\mathbb{E}\left[e^{\lambda \,\mathrm{Bin}(t,p)}\right] = \mathbb{E}\left[e^{\lambda \sum_{s=1}^{t} X_s}\right] = \mathbb{E}\left[\prod_{s=1}^{t} e^{\lambda X_s}\right] = \left(\mathbb{E}\left[e^{\lambda \,\mathrm{Bern}(p)}\right]\right)^t$$

$$= \left(1 - p + pe^{\lambda}\right)^t,$$

where we used independence in the last equality of the first line.

We choose $\lambda = \log\left(\alpha(tp)^{-1}\right)$, assuming $\alpha > tp$, so that $\lambda > 0$. This leads to

$$\mathbb{P}\left(\mathrm{Bin}(t,p) \geq \alpha\right) \leq e^{-\log\left(\frac{\alpha}{tp}\right)\alpha}\left(1 - p + \frac{\alpha}{t}\right)^t \leq \left(\frac{etp}{\alpha}\right)^\alpha. \tag{43}$$

Therefore, the final inequality yields

$$\mathbb{P}\left(x_i^t \in B(v_i, \beta^{-\frac{1}{4}})\right) \geq 1 - \left(\frac{etp}{\left\lfloor \frac{\beta^{-\frac{1}{4}}}{\gamma\,\mathsf{d}(\mathcal{K})} + 1\right\rfloor}\right)^{\left\lfloor \frac{\beta^{-\frac{1}{4}}}{\gamma\,\mathsf{d}(\mathcal{K})} + 1\right\rfloor}$$

for $(t,i) \in [\![0, T_1]\!] \times [\![1, n]\!]$, whenever $\left\lfloor \beta^{-\frac{1}{4}}(\gamma\mathsf{d}(\mathcal{K}))^{-1}\right\rfloor + 1 > pt$.

## B.8. Proof of Theorem 5.4

We henceforth denote

$$\mathfrak{B}_\ell(\varepsilon) := \mathrm{conv}\left\{x_{i\ell}^0\right\}_{i \in [\![1, \mu_\ell]\!]} + B(0, \varepsilon).$$

We begin with the following crucial fact.

**Claim B.8.** *Conditioning on the event*

$$\left\{x_{i\ell}^t \in \mathfrak{B}_\ell(\varepsilon) \text{ for all } (\ell, i) \in [\![1, \kappa]\!] \times [\![1, \mu_\ell]\!]\right\},$$

*where, $\mu_\ell \in [\![1, n]\!]$ denotes the number of points in the cell $\mathscr{C}_\ell(v)$, fix $\ell \in [\![1, \kappa]\!]$ and $i \in [\![1, \mu_\ell]\!]$. We have*

$$p_{(i\ell)\to\ell}^t := \sum_{j=1}^{\mu_\ell} \frac{e^{\beta\langle x_{i\ell}^t, x_{j\ell}^t\rangle}}{\sum_{m=1}^{\kappa}\sum_{k=1}^{\mu_m} e^{\beta\langle x_{i\ell}^t, x_{km}^t\rangle}} \geq 1 - (n - \mu_\ell)\,\mu_\ell e^{-\lambda\beta},$$

*where $\lambda = c_0 - 4\mathsf{d}(\mathcal{K})(\varepsilon + \tau) - 2(\varepsilon + \tau)^2$.*

*Proof of Claim B.8.* First, note that

$$\langle x_{i\ell}^t, x_{j\ell}^t\rangle \geq \langle v_\ell, v_\ell\rangle - 2\mathsf{d}(\mathcal{K})(\varepsilon + \tau) - (\varepsilon + \tau)^2.$$

Hence,

$$\frac{e^{-\beta\left(\langle v_\ell, v_\ell\rangle - 2\mathsf{d}(\mathcal{K})(\varepsilon+\tau) - (\varepsilon+\tau)^2\right)}}{e^{-\beta\left(\langle v_\ell, v_\ell\rangle - 2\mathsf{d}(\mathcal{K})(\varepsilon+\tau) - (\varepsilon+\tau)^2\right)}} \sum_{j=1}^{\mu_\ell} \frac{e^{\beta\langle x_{i\ell}^t, x_{j\ell}^t\rangle}}{\sum_{m=1}^{\kappa}\sum_{k=1}^{\mu_m} e^{\beta\langle x_{i\ell}^t, x_{km}^t\rangle}}$$

$$= \sum_{j=1}^{\mu_\ell} \frac{e^{\beta\langle x_{i\ell}^t, x_{j\ell}^t\rangle - \beta\left(\langle v_\ell, v_\ell\rangle - 2\mathsf{d}(\mathcal{K})(\varepsilon+\tau) - (\varepsilon+\tau)^2\right)}}{\mathfrak{T}} \tag{44}$$

where

$$\mathfrak{L} := \sum_{k=1}^{\mu_\ell} e^{\beta \langle x_{i\ell}^t, x_{k\ell}^t \rangle - \beta \left( \langle v_\ell, v_\ell \rangle - 2\mathsf{d}(\mathcal{K})(\varepsilon + \tau) - (\varepsilon + \tau)^2 \right)}$$
$$+ \sum_{\substack{m=1 \\ m \neq \ell}}^{\kappa} \sum_{k=1}^{\mu_m} e^{\beta \langle x_{i\ell}^t, x_{km}^t \rangle - \beta \left( \langle v_\ell, v_\ell \rangle - 2\mathsf{d}(\mathcal{K})(\varepsilon + \tau) - (\varepsilon + \tau)^2 \right)}.$$

Since $\langle v_i, v_i \rangle \geq \langle v_i, v_\ell \rangle + c_0$ with $c_0 > 0$ for any $\ell \neq i$, we have $x_{i\ell}^t = v_\ell + y_{i\ell}^t$ with $\|y_{i\ell}^t\| \leq \varepsilon + \tau$, and similarly $x_{km}^t = v_m + y_{km}^t$ with $\|y_{km}^t\| \leq \varepsilon + \tau$. Hence,

$$\langle x_{i\ell}^t, x_{km}^t \rangle = \langle v_\ell, v_m \rangle + \langle y_{i\ell}^t, v_m \rangle + \langle y_{km}^t, v_\ell \rangle + \langle y_{i\ell}^t, y_{km}^t \rangle$$
$$\leq \langle v_\ell, v_m \rangle + 2\mathsf{d}(\mathcal{K})(\varepsilon + \tau) + (\varepsilon + \tau)^2$$
$$\leq \langle v_\ell, v_\ell \rangle + 2\mathsf{d}(\mathcal{K})(\varepsilon + \tau) + (\varepsilon + \tau)^2 - c_0.$$

Therefore,

$$\langle x_{i\ell}^t, x_{km}^t \rangle - \langle v_\ell, v_\ell \rangle - 2\mathsf{d}(\mathcal{K})(\varepsilon + \tau) - (\varepsilon + \tau)^2 \leq -c_0, \quad \text{for } m \neq \ell.$$

Consequently, we can lower bound (44) by

$$\sum_{j=1}^{\mu_\ell} \frac{e^{\beta \langle x_{i\ell}^t, x_{j\ell}^t \rangle - \beta \left( \langle v_\ell, v_\ell \rangle - 2\mathsf{d}(\mathcal{K})(\varepsilon + \tau) - (\varepsilon + \tau)^2 \right)}}{\displaystyle\sum_{k=1}^{\mu_\ell} e^{\beta \langle x_{i\ell}^t, x_{k\ell}^t \rangle - \beta \left( \langle v_\ell, v_\ell \rangle - 2\mathsf{d}(\mathcal{K})(\varepsilon + \tau) - (\varepsilon + \tau)^2 \right)} + (n - \mu_\ell)e^{-\beta c_0}}$$

$$\geq \left( 1 + \frac{(n - \mu_\ell)e^{-c_0 \beta}}{\displaystyle\sum_{k=1}^{\mu_\ell} e^{\beta \langle x_{i\ell}^t, x_{k\ell}^t \rangle - \beta \left( \langle v_\ell, v_\ell \rangle - 2\mathsf{d}(\mathcal{K})(\varepsilon + \tau) - (\varepsilon + \tau)^2 \right)}} \right)^{-1}$$

$$\geq 1 - \frac{(n - \mu_\ell)e^{-c_0 \beta}}{\displaystyle\sum_{k=1}^{\mu_\ell} e^{\beta \langle x_{i\ell}^t, x_{k\ell}^t \rangle - \beta \left( \langle v_\ell, v_\ell \rangle - 2\mathsf{d}(\mathcal{K})(\varepsilon + \tau) - (\varepsilon + \tau)^2 \right)}}.$$

We finally bound

$$\langle x_{i\ell}^t, x_{k\ell}^t \rangle - \left( \langle v_\ell, v_\ell \rangle - 2\mathsf{d}(\mathcal{K})(\varepsilon + \tau) - (\varepsilon + \tau)^2 \right) \geq -4\mathsf{d}(\mathcal{K})(\varepsilon + \tau) - 2(\varepsilon + \tau)^2,$$

which yields

$$p_{(i\ell) \to \ell}^t \geq 1 - \frac{(n - \mu_\ell)e^{-c_0 \beta}}{\displaystyle\sum_{k=1}^{\mu_\ell} e^{\beta \langle x_{i\ell}^t, x_{k\ell}^t \rangle - \beta \left( \langle v_\ell, v_\ell \rangle - 2\mathsf{d}(\mathcal{K})(\varepsilon + \tau) - (\varepsilon + \tau)^2 \right)}}$$
$$\geq 1 - (n - \mu_\ell)\mu_\ell \exp \left( -\beta \left( c_0 - 4\mathsf{d}(\mathcal{K})(\varepsilon + \tau) - 2(\varepsilon + \tau)^2 \right) \right). \qquad \square$$

Proceeding similarly to Steps 1 and 2 of Appendix B.7, we deduce that, with probability 1,

$$T_2 \geq \left\lfloor \frac{\varepsilon}{\gamma \mathsf{d}(\mathcal{K})} \right\rfloor.$$

Fix $t_0 := \left\lfloor \varepsilon(\gamma \mathsf{d}(\mathcal{K}))^{-1} \right\rfloor$. Then,

$$\mathbb{P}(T_2 \geq t_0) \geq \mathbb{P} \left( \max_{\substack{i \in [\![1, \mu_\ell]\!] \\ \ell \in [\![1, \kappa]\!]}} M_{i\ell}^{t_0} \leq \frac{\varepsilon}{\gamma \mathsf{d}(\mathcal{K})} \right) = 1.$$

(Recall the definition of $M_{i\ell}^t$ in (22).) Fix $\alpha_0 := \lfloor \varepsilon(2\gamma\mathsf{d}(\mathcal{K}))^{-1} \rfloor$, and recall that $\varepsilon(\gamma\mathsf{d}(\mathcal{K}))^{-1} \geq 2$. We estimate

$$
\mathbb{P}\left( \underbrace{\max_{\substack{i\in[\![1,\mu_\ell]\!] \\ \ell\in[\![1,\kappa]\!]}} M_{i\ell}^{t_0} \leq \alpha_0}_{:=\Omega_0} \right) = \mathbb{P}\left( \bigcap_{\ell\in[\![1,\kappa]\!]} \bigcap_{i\in[\![1,\mu_\ell]\!]} \{M_{i\ell}^{t_0} \leq \alpha_0\} \right) \tag{45}
$$

$$
= 1 - \mathbb{P}\left( \bigcup_{\ell\in[\![1,\kappa]\!]} \bigcup_{i\in[\![1,\mu_\ell]\!]} \{M_{i\ell}^{t_0} > \alpha_0\} \right)
$$

$$
\geq 1 - \sum_{\ell\in[\![1,\kappa]\!]} \sum_{i\in[\![1,\mu_\ell]\!]} \left(1 - \mathbb{P}\left(M_{i\ell}^{t_0} \leq \alpha_0\right)\right).
$$

Note that by the definition of $t_0$, we have

$$
\mathbb{P}\left( \bigcap_{\ell\in[\![1,\kappa]\!]} \bigcap_{i\in[\![1,\mu_\ell]\!]} \{x_{i\ell}^{t_0} \in \mathfrak{B}_\ell(\varepsilon)\} \right) = 1.
$$

Therefore, by Claim B.4, and arguing as in Step 2 of the proof of Theorem 5.2, we obtain (using a Chernoff-type bound (43)):

$$
\mathbb{P}\left(M_{i\ell}^{t_0} \leq \alpha_0\right) \geq 1 - \left(\frac{et_0 p}{\alpha_0}\right)^{\alpha_0} =: 1 - \xi_0
$$

where $p$ is the upper bound of $1 - p_{(i\ell)\to\ell}^t$ in Claim B.8. Therefore, going back to (45), we find

$$
\mathbb{P}\left( \max_{\substack{i\in[\![1,\mu_\ell]\!] \\ \ell\in[\![1,\kappa]\!]}} M_{i\ell}^{t_0} \leq \alpha_0 \right) \geq 1 - n\xi_0 =: \eta_0. \tag{46}
$$

It follows that

$$
\mathbb{P}\left( \left\{ \max_{\substack{i\in[\![1,\mu_\ell]\!] \\ \ell\in[\![1,\kappa]\!]}} \mathrm{dist}\left(x_{i\ell}^{t_0}, \partial\mathfrak{B}_\ell(\varepsilon)\right) \leq \varepsilon - \alpha_0\gamma\mathsf{d}(\mathcal{K}) \right\} \right) \geq \eta_0.
$$

Now, set $t_1 := \lfloor \varepsilon(2\gamma\mathsf{d}(\mathcal{K}))^{-1} \rfloor$. Then,

$$
\mathbb{P}\left( \bigcap_{\ell\in[\![1,\kappa]\!]} \bigcap_{i\in[\![1,\mu_\ell]\!]} \{x_{i\ell}^{t_0+t_1} \in \mathfrak{B}_\ell(\varepsilon)\} \,\Big|\, \Omega_0 \right) = 1.
$$

Much like what is done for (46), we find

$$
\mathbb{P}\left(T_2 \geq t_0 + t_1 \,|\, \Omega_0\right)
$$

$$
\geq \mathbb{P}\left( \max_{\substack{i\in[\![1,\mu_\ell]\!] \\ \ell\in[\![1,\kappa]\!]}} M_{i\ell}^{t_0+t_1} \leq \alpha_0 \,\Bigg|\, \bigcap_{\ell\in[\![1,\kappa]\!]} \bigcap_{i\in[\![1,\mu_\ell]\!]} \{x_{i\ell}^{t_0+t_1} \in \mathfrak{B}_\ell(\varepsilon)\} \right)
$$

$$
\geq 1 - n\left(\frac{(t_0+t_1)p}{\alpha_0}\right)^{\alpha_0} =: 1 - n\xi_1 =: \eta_1.
$$

Therefore,

$$
\mathbb{P}\left( \underbrace{\max_{\substack{i\in[\![1,\mu_\ell]\!] \\ \ell\in[\![1,\kappa]\!]}} M_{i\ell}^{t_0+t_1} \leq \alpha_0}_{:=\Omega_1} \right) \geq \eta_0\eta_1.
$$

This again implies

$$\mathbb{P}\left(\bigcap_{\ell\in[\![1,\kappa]\!]}\bigcap_{i\in[\![1,\mu_\ell]\!]}\left\{x_{i\ell}^{t_0+2t_1}\in\mathfrak{B}_\ell(\varepsilon)\right\}\,\Big|\,\Omega_1\right)=1.$$

We can repeat this procedure to obtain

$$\mathbb{P}\left(T_2\geq t_0+Lt_1\right)\geq\mathbb{P}\left(\max_{\substack{i\in[\![1,\mu_\ell]\!]\\\ell\in[\![1,\kappa]\!]}}M_{i\ell}^{t_0+Lt_1}\leq\alpha_0\right)\geq\prod_{j=0}^{L}\eta_j,$$

where

$$\eta_j=1-n\left(\frac{(t_0+jt_1)p}{\alpha_0}\right)^{\alpha_0}.$$

Now fix $t>1$. Recalling that $t_1=\left\lfloor\varepsilon(2\gamma\mathsf{d}(\mathcal{K}))^{-1}\right\rfloor$ and $t_0=\left\lfloor\varepsilon(\gamma\mathsf{d}(\mathcal{K}))^{-1}\right\rfloor$, we choose the smallest integer $L\geq 1$ such that

$$t\leq t_0+Lt_1,$$

namely

$$L=\left\lceil\frac{t-\left\lfloor\frac{\varepsilon}{\gamma\mathsf{d}(\mathcal{K})}\right\rfloor}{\left\lfloor\frac{\varepsilon}{2\gamma\mathsf{d}(\mathcal{K})}\right\rfloor}\right\rceil.$$

To estimate the final probability, we upper bound $L$ by

$$L\leq\frac{2\gamma\mathsf{d}(\mathcal{K})}{\varepsilon}t.$$

Hence,

$$\begin{aligned}
\prod_{j=0}^{L}\eta_j&=\prod_{j=0}^{L}\left(1-n\left(\frac{(t_0+jt_1)p}{\alpha_0}\right)^{\alpha_0}\right)\\
&\geq\left(1-n\left(\frac{(t_0+Lt_1)p}{\alpha_0}\right)^{\alpha_0}\right)^{L}\\
&\geq 1-Ln\left(\frac{(t_0+Lt_1)p}{\alpha_0}\right)^{\alpha_0}+O\left(\left(n\left(\frac{(t_0+Lt_1)p}{\alpha_0}\right)^{\alpha_0}\right)^{2}\right)\\
&=1-\exp\left(\log L+\log n+\alpha_0\log\left(\frac{t_0+Lt_1}{\alpha_0}\right)+\alpha_0\log p\right)+O(f(p)^2),
\end{aligned}$$

where

$$f(p):=n\left(\frac{(t_0+Lt_1)p}{\alpha_0}\right)^{\alpha_0}.$$

All in all,

$$\mathbb{P}(T_2\geq t)\geq 1-\exp\left(\log L+\log n+\alpha_0\log\left(\frac{t_0+Lt_1}{\alpha_0}\right)+\alpha_0\log p+O(1)\right)$$

where $p$ is the upper bound of $1-p_{(i\ell)\to\ell}^t$ in Claim B.8. This concludes the proof. $\qquad\square$

