# OpenReview forum: "Attention's forward pass and Frank-Wolfe"
_ICML.cc/2026/Conference — ICML 2026 regular_

### Official Review · Reviewer_P8KK · 2026-02-24

**Soundness:** 3
**Presentation:** 3
**Significance:** 3
**Originality:** 4
**Overall Recommendation:** 4
**Confidence:** 4

**Summary:**

This paper provides a rigorous mathematical analysis of how token embeddings evolve through the layers of a Transformer during inference, focusing on the self-attention mechanism. The authors study the so-called limit of self-attention, which arises when the inverse-temperature parameter $\beta \to +\infty$, and show that in this regime the update rule for each token can be interpreted as a Frank--Wolfe optimization step over the convex hull of the current token embeddings. Depending on the sign of the key-query matrix, two distinct behaviors emerge: when the matrix is negative semidefinite, tokens cluster toward a common point at a sublinear $\mathcal{O}(t^{-1})$ rate; when it is positive semidefinite, the convex hull induces a Voronoi-like partition in which vertices remain fixed and interior tokens converge exponentially fast along straight lines toward the vertex of their respective cell. The authors additionally establish well-posedness of the associated continuous-time ODE, partially resolving an open problem in the prior literature. For the practically relevant case of finite $\beta$, the self-attention process is modeled as a Markov chain, and the authors prove \textit{dynamic metastability}: tokens rapidly cluster near vertex configurations in a constant number of steps and remain trapped there for a time that grows exponentially in $\beta$, before eventually collapsing to a single point. This result formally justifies using the hardmax dynamics as an accurate approximation of softmax self-attention over exponentially long time horizons.

**Compliance With Llm Reviewing Policy:**

Affirmed.

**Key Questions For Authors:**

Question 1:
Your analysis requires the key-query matrix to be symmetric, but in practice it is the product of two different matrices and is generally not symmetric. Do you expect the Frank-Wolfe interpretation and convergence results to survive in the non-symmetric case, or does the geometric picture break down entirely?

Question 2:
Regarding the vertex-position assumptions in Theorem 4.2. , the convergence result requires each vertex to belong strictly to its own Voronoi cell. Do you have any evidence, even on simple synthetic token distributions, that this condition holds approximately in settings resembling real Transformers?

Question 3:
Many of your results, such as straight-line convergence to vertices and exponential metastability timescales, are concrete enough to verify numerically. Have you tried any simple simulations, and if so, do the theoretical predictions hold up?

Question 4:
Your model omits MLP blocks, uses a single head, and replaces layer normalization with a simpler Euclidean renormalization. Which of these simplifications do you consider most consequential, and do you expect the core insights to carry over to more realistic architectures?

Question 5:
In paper theorem 5.2 , assumes equal vertex norms and a strict separation condition. How sensitive are the exponential trapping times to violations of these conditions, and does moderate irregularity in the geometry significantly shorten the metastable window?

**Limitations:**

The authors briefly acknowledge the main technical limitations of their work, specifically the restriction to symmetric key-query matrices and the open problem of removing the vertex-position assumptions, and they note these as directions for future work. However, the discussion of limitations is somewhat minimal and scattered across the paper rather than consolidated in one place. It would be helpful if the authors added a short dedicated paragraph that brings together all the key limitations, including the single-head encoder-only setup, the absence of MLP components, the simplified renormalization scheme, and the restrictive geometric assumptions required for the metastability results, so that readers can quickly gauge the boundaries of the theoretical framework. On the question of societal impact, the authors include a brief statement noting that they see no specific negative societal consequences of their work, which is reasonable given the purely theoretical nature of the contribution. Overall, the limitations are acknowledged but could be discussed more systematically and in much greater depth.

**Strengths And Weaknesses:**

The paper is mathematically solid and the main results are backed by complete, carefully written proofs. The authors are upfront about what their analysis cannot yet handle, such as non-symmetric key-query matrices and the restrictive assumptions on vertex positions, which is a sign of intellectual honesty. That said, the paper would have felt more grounded if the authors had included even a few simple simulations to show that the theoretical predictions actually match what happens in practice. The writing is clear and the paper flows well, taking the reader step by step from the hardmax limit to the Frank-Wolfe interpretation and then to the metastability result. The proofs are kept in the appendix so the main text stays readable, which is a good choice. The one place where the paper loses some readers is in the discussion of the Voronoi cell geometry, which could use a more intuitive explanation for those less familiar with the topic. In terms of its broader importance, the paper tackles a question that genuinely matters, namely how information moves through a Transformer during inference, and the finding that hardmax dynamics closely approximate softmax attention for an exponentially long time is both surprising and useful. It helps explain empirical observations like attention sparsity and token clustering that practitioners have noticed but not fully understood. The core idea of viewing self-attention as a Frank-Wolfe step is fresh and clever, and the supporting results on Voronoi structure, ODE well-posedness, and metastability all add real value. Overall, this is a strong theoretical paper that opens up new ways of thinking about how Transformers work.

---

> ### Author Rebuttal · Authors · 2026-03-30
>
> We thank the reviewer for carefully reading our submission and for their comments. We address them below.
>
> ## Strengths and Weaknesses
>
> > [...] the paper would have felt more grounded if the authors had included even a few simple simulations to show that the theoretical predictions actually match what happens in practice.
>
> We thank the reviewer for their suggestion and have included simulations of the finite-$\beta$ model. We refer to **Key Questions for Authors, Question 3** for all details.
>
> ## Key Questions for Authors
> 1. We refer to the response to **Reviewer DCFY, Key Question 2** for a detailed answer.
>
> 2. We thank the reviewer for this question. While we do not test this condition directly on realistic transformer embeddings, we provide theoretical evidence in Remarks 4.7 and 4.8 that it holds in natural regimes. In particular, for Gaussian vertices in high dimension with well-conditioned $B$, the condition holds with high probability. We also characterize it geometrically via an angle condition, which can be interpreted as a local optimality property of the vertices. These results suggest that the assumption is reasonable in high-dimensional or well-separated settings.
>
>    Furthermore, the results are almost unaltered if the vertices do not belong to their own cell. In a first phase, all points will cluster near local maxima, which are invariant for the dynamics and are a subset of the vertices. Then in a second phase, there will be an eventual collapse. The main difference is the difficulty to quantify which tokens go to which local-maxima vertices and what is the relative mass of each cluster in the metastable state, hence complicating the technical part of the analysis.
>
> 3. We thank the reviewer for this suggestion. We performed numerical simulations in a simple setting (e.g., two dimensional particles and $\beta = 10$) and observed strong qualitative agreement with our theoretical predictions.
>
>    In particular, we see an initial rapid convergence toward vertex configurations, followed by a metastable regime (around $t \approx 200$) where the system remains close to these configurations for a long time. Final collapse occurs much later (around $t \approx 3000$), consistent with the predicted separation of time scales.
>
>    A minor deviation is that some nearby vertices merge during the early phase, which we attribute to not enforcing the precise initialization assumptions required by the theory in Section 5. Nevertheless, the overall dynamical picture (fast clustering, long metastability, and delayed collapse after a time exponential in $\beta$) is clearly observed (see Fig. beta_case in the supplementary material: https://github.com/submission8097-dev/Submission8097).
>
> 4. We thank the reviewer for this question. While we expect our insights to qualitatively carry over to more realistic architectures, our goal is to isolate and understand the geometric mechanisms induced by self-attention. Thus, we view our analysis as providing tools and intuition for studying more complex settings, rather than a complete description of them.
>
>    As evidence that our insights should carry over to more realistic transformers, we point to the simulations discussed in our response to **Reviewer DCFY, Strengths and Weaknesses**, where including simple MLP layers preserves a similar qualitative behavior: rapid clustering towards vertices, metastability, and delayed collapse, albeit with geometric deformations of the initial cell geometry.
>
>    Regarding normalization, our Euclidean renormalization is a proxy for controlling scale. Alternative idealizations, such as projection onto the sphere (as in [Geshkovski et al., 2024] (https://arxiv.org/abs/2410.06833)), also exhibit similar metastable behavior.
>
> 5. We expect the order of $\mathcal{O}(e^\beta)$ to be maintained for the metastable window as long as the geometry is moderately perturbed. Partial numerical evidence of this fact is provided in the numerical simulations included in **Reviewer DCFY, Strengths and Weaknesses**. Here, the added MLP residual block deforming the cell geometry does not alter the emergence of a metastable window nor does it shorten it significantly.
>
> ## Limitations
> We thank the reviewer for this helpful suggestion and agree that a clearer and more systematic presentation of the limitations would improve the manuscript. In the revised version, we will include a  paragraph in Section 6 consolidating the main modelling assumptions and limitations of our framework.

---

> > ### Author Rebuttal · Reviewer_P8KK · 2026-04-01
> >
> > The authors have addressed several of my concerns, and I appreciate the added simulations and the clarification on the role of MLP layers. The simulation evidence showing that the qualitative behavior (rapid clustering, metastability, delayed collapse) persists with ReLU feedforward layers is reassuring and partially addresses my concern about practical relevance.
> >
> > However, two concerns remain open for me.
> >
> > First, on the question of practical relevance: the authors argue that collapse occurs only after exponentially many layers, and therefore practical transformers avoid this regime. I find this argument plausible but not fully convincing. Real transformers are trained, not just run at inference with fixed weights. During training, the weight matrices are updated continuously, and it is not clear that the metastability argument applies in the training regime. Could the authors clarify whether their analysis gives any insight into how the identified parameter regimes (those avoiding degenerate collapse) relate to what gradient descent actually finds during training?
> >
> > Second, on the non-symmetric key-query matrix: the authors explain that the asymmetric part induces a rotational drift and that the interplay between alignment (from the symmetric part) and rotation (from the antisymmetric part) remains open. I appreciate the honesty. My question is whether there are any parameter regimes, even simple ones such as small antisymmetric perturbations, where the metastability picture provably or numerically survives. Even a simple example would help the reader gauge how fragile the symmetric assumption is.

---

> > > ### Author Response · Authors · 2026-04-06
> > >
> > > We thank the reviewer for their questions. We respond to them below.
> > >
> > > > First, on the question of practical relevance: the authors argue that collapse occurs only after exponentially many layers, and therefore practical transformers avoid this regime. I find this argument plausible but not fully convincing. Real transformers are trained, not just run at inference with fixed weights. During training, the weight matrices are updated continuously, and it is not clear that the metastability argument applies in the training regime. Could the authors clarify whether their analysis gives any insight into how the identified parameter regimes (those avoiding degenerate collapse) relate to what gradient descent actually finds during training?
> > >
> > > Our work focuses on forward dynamics, i.e., the behavior of token representations under a fixed set of trained parameters. As such, we do not model or analyze the training dynamics, and cannot make direct claims about which configurations are selected by gradient-based optimization.
> > >
> > > That said, we believe our results can still provide useful *inductive biases* for model design and parametrization. In particular, our analysis identifies regimes that lead to degenerate behavior (such as collapse to a single point in the negative semi-definite case, cf. Theorem 3.1), and others that exhibit more structured dynamics (clustering and metastability in the positive definite case). This suggests that, if one wishes to avoid undesirable regimes during training, one could incorporate structural constraints at the level of parametrization. For instance, to exclude the negative semi-definite regime, one could parametrize
> > > $$
> > > B^t = (\tilde{B}^t)^\top \tilde{B}^t
> > > $$
> > > thereby enforcing positive semi-definiteness of $B^t$ throughout training and ensuring that the forward dynamics remain in the regime analyzed in our work.
> > >
> > > Finally, we note that the metastable state is reached *fast* (with few layers) while the full collapse to a point is *slow* (exponentially many layers). For this reason, for a given time-dependent matrix $B^t$ (from a trained transformer) that varies slowly across some layers, a metastable state analogous to those studied in the paper could emerge. However, the precise quantification of such a phenomenon with matrices depending on time remains open.
> > >
> > >
> > > > Second, on the non-symmetric key-query matrix: the authors explain that the asymmetric part induces a rotational drift and that the interplay between alignment (from the symmetric part) and rotation (from the antisymmetric part) remains open. I appreciate the honesty. My question is whether there are any parameter regimes, even simple ones such as small antisymmetric perturbations, where the metastability picture provably or numerically survives. Even a simple example would help the reader gauge how fragile the symmetric assumption is.
> > >
> > > As suggested by the reviewer, one expects small non-symmetric perturbations of $B$ to preserve the qualitative features predicted by the symmetric analysis. Our numerical experiments (with $B$ being the identity matrix plus a non-symmetric perturbation) support this. When the skew-symmetric component has a small norm ($\|K\|_F = 0.1$), the dynamics and geometric structure remain close to the symmetric case (see Fig. small_nonsym_case in https://github.com/submission8097-dev/Submission8097). In contrast, when the perturbation is larger ($\|K\|_F = 1$), two main deviations appear: the induced cell partition becomes geometrically distorted, and rotational effects dominate, leading to significantly faster collapse ($t\approx 10$), and a marked reduction of the metastable regime (see Fig. large_nonsym_case in https://github.com/submission8097-dev/Submission8097).

---

### Official Review · Reviewer_SzHQ · 2026-03-11

**Soundness:** 4
**Presentation:** 4
**Significance:** 3
**Originality:** 3
**Overall Recommendation:** 5
**Confidence:** 3

**Summary:**

This paper seeks to understand the self-attention mechanism in the limit as the inverse temperature parameter $\beta \to \infty$. Within this regime, the layer by layer changes which are applied to the input sequence can be seen as an update rule over a quadratic potential in a convex subspace of the embedding space. Within this light, the authors view the process as a series of Frank-Wolfe updates with the value matrix acting as a preconditioner. The ensuing dynamics converge towards cells which tesselate the underlying space in line with a Voronoi structure. They then apply these insights to the finite-temperature regime via a Gumbel transformation, demonstrating dynamic metastability.

**Compliance With Llm Reviewing Policy:**

Affirmed.

**Final Justification:**

This work is strong and important to the community at ICML. My primary concerns with this work dealt with the underlying model, namely the assumptions on the matrices $B_t$ and value matrices $V_t$. The responses of the authors addressed these concerns with experimental studies of the behavior beyond this regime and clarifying they study the symmetric part of the dynamics. With this in mind, I raise my score.

**Key Questions For Authors:**

>- How do the dynamics change when we assume non-trivial structure over the matrix $V_t$?
>- How exactly does the metastability in this system persist? Is there cut-off phenomena, where the chain moves rapidly in a short amount of time? Is there rapid movement within each Voronoi cell, e.g. fast local mixing?

**Limitations:**

yes

**Strengths And Weaknesses:**

**Strengths**
>- The paper is well-written and theoretically sound with clear objectives and an easy to understand narrative.
> - Demystifying the self-attention mechanism is an important question to the machine learning community, and analysis of the zero temperature limit is a natural manner to gain theoretical insights on its behavior.
> - The connection to Frank-Wolfe updates and methods in the optimization literature is natural and provides additional ways of understanding the attention mechanism.
> - The paper reaches a satisfying conclusion of theoretical interest, which is that the hardmax dynamics of a simplified self-attention mechanism follow a well-known optimization framework, and have an intricate geometric interpretation.

**Weaknesses**
>- As a theoretical paper, it is natural to make assumptions over the model to simplify the analysis, yet the assumption that the matrix $B_t$ is static for all $t$ and positive semi-definite is quite strong, and unrealistic as it is unlikely the product of two matrices $W_K$ and $W_Q$ will achieve this.
>- Moreover, the authors assume the value matrix, the pre-conditioner $V_t$ is a multiple of the identity matrix. As an integral part of the attention mechanism, it is interesting to explore how the dynamics would change if this matrix had non-trivial structure.
>- Additional discussion of metastability in the finite-temperature, and implication for learning would be appreciated.

---

> ### Author Rebuttal · Authors · 2026-03-30
>
> We thank the reviewer for their questions, which we address below.
>
> ## Strengths and Weaknesses
> ### Weaknesses
> - When $B^t$ varies with $t$, the Frank-Wolfe objective $J$ changes across layers. We can expect similar behavior under slow variation conditions, while rapid changes may obscure the geometric interpretation. In such cases, one can group layers into regimes of slow variation and apply our analysis within each group.
>
>   Regarding the semi-definiteness assumptions, note that we analyze the single-head case ($H = 1$), where positive definiteness is certainly feasible. In the multi-head setting ($H > 1$), matrices take the form $B_h^t = (Q_h^t)^\top K_h^t$ with $Q_h^t, K_h^t \in \mathbb{R}^{d/H \times d}$, implying the presence of zero eigenvalues. Our results for semi-definite matrices extend to this setting, but not those requiring strict definiteness (as the metastability results of Section 5). We will clarify this point in the revised Section 5.
>
> - The assumption $V^t = \gamma^t I_d$ allows us to isolate and analyze the geometric effects of self-attention in a clean setting, but we agree that understanding more general value matrices is of clear interest, as a key component of self-attention layers.
>
>   When $V^t$ has non-trivial structure, it alters the trajectories of the embeddings by changing how information is propagated across layers. In particular, the vertices are no longer constrained to evolve along fixed directions, but instead follow dynamics shaped by $V^t$, which can introduce additional effects such as rotations or deformations.
>
>   To investigate this, we performed simulations in two dimensions where $V^t$ is chosen as a rotation matrix. In this case, the dynamics exhibit an additional rotational component, with both particles and vertices undergoing rotation-like motion. Despite this modification, the overall behavior remains consistent with our theory: we still observe rapid clustering toward the vertices, followed by metastability and eventual collapse (see Fig. V_case in the supplementary material: https://github.com/submission8097-dev/Submission8097).
>
>   These experiments indicate that the phenomena we describe are robust to such changes in $V^t$. Extending the theoretical analysis beyond the positive scalar case would require adapting our tools to track the induced geometry on a case-by-case basis, which we leave for future work.
>
> - According to our interpretation, the metastability property is what makes self-attention dynamics compatible with practical transformer architectures. While self-attention induces clustering and eventual full collapse, metastability separates these effects in time: partial clustering occurs early and can support mixing, whereas full collapse is delayed.
>
>   Our results show that, at finite temperature, full collapse happens only after exponentially many layers. This suggests that practical transformers operate in a regime where useful clustering that promotes mixing is present, but collapse lies beyond the model depth.
>
>   Finally, although our analysis concerns forward dynamics, it points to parameter regimes and scalings that should be avoided during training to prevent degenerate behavior.
>
> ## Key Questions for Authors
> - We point to the response in Strengths and Weaknesses above.
>
> - One of the benefits of the analysis presented in Section 5 is that it provides a clear picture of how the metastability emerges and with which timescales (as a function of $\beta$). As the reviewer correctly points out, and as discussed in the last bullet point of Section 1.2, our results show how self-attention dynamics exhibit dynamic metastability: with high probability, particles rapidly approach near-vertex configurations (Theorem 5.2) and remain trapped for times exponential in $\beta$ (Theorem 5.4), after which they eventually collapse to a single cluster (Proposition 5.1).

---

> > ### Author Rebuttal · Reviewer_SzHQ · 2026-04-02
> >
> > This work is strong and outlines the dynamics for self-attention at all temperatures, when the composite key-query matrix is positive semi-definite. However some minor issues remain. The practical interpretation of the dynamics in light of transformers is a bit lacking, i.e. what does collapse mean in terms of the understanding of the model, and the PSD assumption is a bit strong as it is unlikely for the product of two matrices to be PSD.
> >
> > These thoughts in mind, I keep my positive score.

---

> > > ### Author Response · Authors · 2026-04-06
> > >
> > > We thank the reviewer for their positive assessment and for the follow-up questions. We address the points raised below.
> > >
> > > ### Practical interpretation of collapse
> > >
> > > In our framework, collapse corresponds to a loss of token diversity: all token embeddings converge to a single point, or a very small set of points. This could be interpreted either as a loss of information or as a form of information compression. Since we consider encoder-type transformers, we find the metastable state as a reasonable mechanism to compress information.
> > >
> > > However, an important point of our analysis is that full collapse into a point (which could be problematic if interpreted as information loss) occurs only after exponentially many layers in $\beta$. What we show is that, before this regime, the system exhibits *metastability*, where tokens group into a small number of coherent clusters, and this occurs after a small number of layers. We interpret this metastable regime as the potentially useful operating regime of transformers.
> > >
> > > ### On the PSD assumption
> > >
> > > We agree with the reviewer that assuming $B = W_Q^\top W_K$ to be PSD is a strong idealization, since in practice it is the product of two independently learned matrices and will typically be indefinite and non-symmetric.
> > >
> > > Our analysis should be understood as a decomposition of regimes *within the symmetric part of the dynamics*. In particular, when $B$ is symmetric, we distinguish:
> > > - the NSD regime, where the Frank–Wolfe structure leads directly to collapse,
> > > - the PD regime, where we obtain clustering to vertices followed by eventual collapse.
> > >
> > > In the general case, one can decompose
> > > $$
> > > B = \tfrac12(B+B^\top) + \tfrac12(B-B^\top) = S + K,
> > > $$
> > > where $S$ is symmetric and typically indefinite, and $K$ is skew-symmetric. Our results describe the dynamics induced by $S$ in its positive and negative spectral components. A natural conjecture is that, in practice, directions corresponding to positive eigenvalues of $S$ exhibit clustering and metastability, while negative directions drive contraction, leading to a combination of the two behaviors. Making this precise remains an open problem.
> > >
> > > Additionally, the skew-symmetric component $K$ is expected to introduce rotational dynamics. We do not analyze this component theoretically, but we expect our description to remain informative when these effects are not dominant.
> > >
> > > To assess the robustness of this picture, we perform numerical simulations with matrices of the form $B = I_d + K$, where $K$ is non-symmetric. When the norm of the non-symmetric part is small ($\|K\|_F = 0.1$), the qualitative dynamics remain close to those of the symmetric case (see Fig. small_nonsym_case in https://github.com/submission8097-dev/Submission8097). As the norm of the non-symmetric part grows larger ($\|K\|_F = 1$), two noticeable differences emerge. First, the cell geometry is deformed, so that cells may contain multiple vertices. Second, rotational effects dominate the dynamics, so collapse happens at a much faster timescale ($t\approx 10$), significantly shortening the metastable regime (see Fig. large_nonsym_case in https://github.com/submission8097-dev/Submission8097).

---

### Official Review · Reviewer_DCFY · 2026-03-13

**Soundness:** 4
**Presentation:** 2
**Significance:** 1
**Originality:** 3
**Overall Recommendation:** 5
**Confidence:** 3

**Summary:**

The authors study signal propagation in a transformer without fully connected layers in the limit of zero temperature and for symmetric key-query product. For most of the analysis there is the additional assumption that the key-query product is the same in each layer. In this limit, moving one layer into the transformer can be seen as a single step of a Frank-Wolfe iteration. This iteration has a geometric interpretation, and the convex hull of the embeddings at each subsequent layer is included in the one of the previous one. If the key-query product is negative definite, the embeddings shrink to a point as they propagate through the network. If it's positive definite, the tokens that are vertices of the hull at the input are fixed, the ones in the interior will get attracted exponentially fast to one of the vertices. The interior of the hull will in fact get divided in cells based on where the points in the interior will converge to. The paper then discusses the case of finite temperature: the tokens in the interior will move to their corresponding vertices in their cells and stay there for a time that scales exponentially with $\beta$, then they will all collapse to some point in the interior.

**Compliance With Llm Reviewing Policy:**

Affirmed.

**Final Justification:**

The authors addressed all my concerns on the practical relevance of this work, showing how the analysis persists even in less idealized settings.  I am happy to raise my score to accept

**Key Questions For Authors:**

1. Around line 28 in the introduction you state that "The study of
signal propagation through the lens of interacting particle
systems has borne fruit" by citing some highly technical document. Could you summarize how exactly the study of signal propagation in the way that is done in this paper can actually help in designing better architectures or training them?

2. How do you expect the results to change if the key-query product is not necessarily symmetric? Do you still expect the embeddings not to collapse for exponentially large number of steps in $\beta$?

3. If I were to summarize the results for finite $\beta$, the embeddings at first collapse onto a set of them (the vertices), and then all with each others in the interior. I would consider both of these stages bad for the network, I would like the embeddings of different tokens to mix while not collapsing. What is preventing this to happen in this model?

**Limitations:**

yes

**Strengths And Weaknesses:**

The paper presents an elegant and technically precise analysis of signal propagation in a simplified model of transformer self-attention. The authors derive a clear geometric interpretation of the token dynamics in the hardmax limit and analyze it with considerable mathematical care. The resulting picture provides an interesting and original perspective on how attention may organize token representations. Overall, the mathematical arguments appear sound and the presentation is generally clear, although the level of rigor and technical detail may limit accessibility for readers who are less interested in the underlying mathematical formalism.

My main reservation concerns the practical relevance of the model. The analysis relies on several strong modeling assumptions: the key–query matrix is assumed to be symmetric, and the architecture omits the feedforward (MLP) layers that are interleaved with attention in real transformers. These modeling choices significantly simplify the dynamics and are essential for the geometric structure derived in the paper. However, it is unclear whether the conclusions remain meaningful once these components are included. In particular, it is difficult to assess whether the convex-hull geometry and cell-based dynamics would survive the presence of nonlinear feedforward layers between attention blocks.

From a practical perspective, the main takeaway seems to be that in this model lower-temperature attention (large $\beta$) can induce metastable clustering behavior and prevent rapid collapse of token embeddings. I am not sure how this translates to real transformer models.

In summary, I view this work as an interesting and original mathematical exploration inspired by self-attention dynamics. However, in its current form the connection to practical transformer architectures appears somewhat tenuous, and additional discussion or empirical evidence supporting the relevance of the model assumptions would strengthen the paper.

---

> ### Author Rebuttal · Authors · 2026-03-30
>
> We thank the reviewer for the careful reading of our manuscript and constructive feedback.
> ## Strengths and Weaknesses
> > [...] In particular, it is difficult to assess whether the convex-hull geometry and cell-based dynamics would survive the presence of nonlinear feedforward layers between attention blocks.
>
> We agree that our simplifying assumptions (symmetric key–query matrix and omission of feedforward layers) are crucial to obtain the clean geometric structure in our analysis.
>
> That said, we believe the main phenomena we identify, namely, convex hull geometry and cell-based dynamics, are not artifacts of these assumptions. When nonlinear feedforward (FF) layers are added, the geometry is deformed, but the qualitative behavior persists: we still observe rapid convergence to extremal points, metastability near them, and eventual collapse.
>
> To support this claim, we ran simulations combining self-attention layers with FF layers of the form $FF(x) = x + w \mathrm{ReLU}(a^\top x + b)$, with fixed parameters $w,a \in \mathbb{R}^2$ and $b\in \mathbb{R}$. While the ReLU FF deforms the geometry (e.g., accelerating merging of nearby vertices in the half space where it is nonzero), the same dynamical picture predicted by our theory appears (see Fig. MLP_case in https://github.com/submission8097-dev/Submission8097).
>
> > [...] additional discussion or empirical evidence supporting the relevance of the model assumptions would strengthen the paper.
>
> We emphasize that our assumption on the value matrix is partially supported by empirical evidence from pretrained encoder models. In [Trockman et al., 2023] (https://arxiv.org/pdf/2305.09828) they observe that, in pretrained vision transformers, value matrices exhibit a strong diagonal structure. The authors further show that this structure can be enforced at initialization (e.g., using identity-like matrices), leading to stable and competitive training. Similarly, [Geshkovski et al., 2025] (https://arxiv.org/pdf/2312.10794) study a simplified model that reproduces phenomena observed in trained architectures such as ALBERT (https://arxiv.org/pdf/1909.11942).
>
> This suggests that analyzing attention dynamics with simple value matrices (e.g., multiples of the identity) captures a regime that is analytically tractable and empirically relevant. We clarify this connection in the revised version.
> ## Key Questions for Authors
> 1. The key contribution of [Cowsik et al., 2024] (https://arxiv.org/abs/2403.02579) is to provide a dynamical systems analysis of how token representations evolve across layers, and to characterize trainability in terms of stability of this evolution. In particular, the paper identifies regimes where signals either collapse or become unstable, and shows that maintaining stable propagation requires appropriately scaling the strength of attention interactions (our $\beta$ parameter) with depth. These results translate into practical guidance: they justify depth-dependent scaling of these interactions at initialization, which has been adopted in large-scale training setups (e.g., OLMo2) to improve stability and performance.
> 2. In the general $B$ non-symmetric setting, we decompose $B = S + K$, where $S = S^\top$ and $K = -K^\top$. Hence, the quadratic form $J(x) = \tfrac{1}{2}\langle Bx, x\rangle = \tfrac{1}{2}\langle S x, x\rangle$ depends only on $S$, so $Bx$ is not a gradient unless $B$ is symmetric.
>
>    Nevertheless, the update retains a Frank-Wolfe-type structure: at each step, we solve the same linear minimization problem using the direction $Bx$, and move toward the solution. This is therefore a conditional gradient scheme with a general (non-gradient) direction; the classical Frank–Wolfe method is recovered when $B$ is symmetric.
>
>    Since convex hull contraction is independent of $B$, we still expect collapse in large time to a single point. The main difficulty in the non-symmetric case is to detect, if there is, a metastable state: $S$ drives alignment, while $K$ induces rotational drift. Understanding their interplay remains open.
>
>    We will include a summarized version of the previous points in the revised version of Section 6.
> 3. Our analysis isolates the intrinsic geometric dynamics induced by self-attention (clustering to vertices, metastability, and eventual collapse) which can persist beyond the simplified setting, as also suggested by simulations including FF layers.
>
>    The key issue is therefore not the complete prevention of collapse, but the time scale over which these dynamics unfold across layers. For finite $\beta$, the system exhibits an initial clustering phase followed by a much slower drift toward full collapse. These time scales depend sensitively on parameters such as $\beta$.
>
>    In particular, since collapse arises only after exponentially many layers, it is unlikely to be reached in realistic settings. Thus, practical models could avoid these dynamics and operate in a regime that balances their effects to avoid degeneracy.

---

> > ### Author Rebuttal · Reviewer_DCFY · 2026-04-03
> >
> > The authors have addressed my concerns. I am happy to raise the score.

---

### Official Review · Reviewer_HF9A · 2026-03-23

**Soundness:** 4
**Presentation:** 4
**Significance:** 4
**Originality:** 4
**Overall Recommendation:** 6
**Confidence:** 5

**Summary:**

This paper studies the zero-temperature (hardmax) limit of a transformer model, and connects that limit to finite-beta behavior. The authors interestingly note that the hardmax update can be interpreted as a Frank-Wolfe step over the convex hull of the current particles. From there, the paper distinguishes two complementary cases according to the sign of the symmetric key-query matrix: in the negative semidefinite case it obtains a $O(t^{-1})$ decay of a quadratic energy, while in the positive definite case it gives a very insightful and much more geometric description in terms of cell decompositions/Voronoi-type regions, stationary vertices, and exponential convergence of interior particles to assigned vertices. The paper then extends the results obtained above to the finite $\beta$ regime, modeling the dynamics as a Markov chain, and proves a detailed picture of the metastable dynamics in which particles first approach near-vertex configurations, then remain trapped there for exponentially long times, before eventual collapse.

**Compliance With Llm Reviewing Policy:**

Affirmed.

**Key Questions For Authors:**

- For the positive-definite regime, the paper gives a very precise geometric picture once the cell structure is fixed. Would the authors comment on what they believe is the main obstruction to removing the vertex/cell assumptions altogether?
- In Section 5, the metastability results are proved under the specialized regime B = Id and fixed gamma. Perhaps the authors could comment on which parts of the argument are genuinely tied to those choices, and which parts are expected to extend to general B positive definite or mildly time-dependent step sizes?

**Limitations:**

yes

**Strengths And Weaknesses:**

This is an excellent paper on a central theoretical question, and it makes a substantial contribution to the mathematical understanding of attention dynamics. Contrarily to much of the present literature, it focuses on the spectral properties of the key-query interaction, as opposed to the one of the value matrix. The resulting analysis is both mathematically clean and conceptually informative: in the negative semidefinite regime, the hardmax dynamics is elegantly tied to Frank-Wolfe, while in the positive definite regime the paper develops a rich and quite elegant geometric description of the finite-particle dynamics, including distinct phases of the evolution and exponential convergence toward selected vertices.
The finite-beta analysis is also particularly valuable, as it rigorously quantifies in what sense the soft dynamics approximates the hardmax limit over long time scales, thereby quantitatively characterizing the metastable behavior of this system that was identified (but still proved quite difficult to study analytically) in the literature. This significantly enhances the practical relevance of the work, since it clarifies how spectral properties of the system parameters shape concentration and collapse phenomena in attention mechanisms.

The paper is very well written, the results are clearly stated and highly original, the contribution is well contextualized and quite significant. It constitutes without a doubt a groundbreaking contribution to the theory of residual streams and information processing in deep transformer models.

---

> ### Author Rebuttal · Authors · 2026-03-30
>
> We thank the reviewer for the positive and thoughtful assessment, and for highlighting both the conceptual and technical contributions of the paper. We respond their questions below.
> ## Key Questions for Authors
> - The vertex assumptions in Section 4 are not a major obstruction to our results and in fact can be removed, at the cost of more technicalities and more difficult characterization of how much mass  each vertex concentrates (in the sense of how many particles have converged to it). We choose to present our results under the vertex assumptions because of the clear picture via the cell geometry that it provides. We discuss this point in more detail in the revised version of Section 6.
> - The restriction to $B=I_d$ in Section 5 is mainly notational: all arguments extend to symmetric positive definite $B$ by replacing the Euclidean inner product with the $B$-inner product, as noted in the footnote of Page 6. Our arguments also extend to time-varying step sizes $\gamma^t$ with minor modifications under the assumption that they remain positive and do not grow too large in time.

---

> > ### Author Rebuttal · Reviewer_HF9A · 2026-04-03
> >
> > The authors fully addressed my questions. I will keep my score unchanged.

---

### Decision · Program_Chairs · 2026-04-30

**Decision:**

Accept (regular)

**Comment:**

The paper studies the dynamics of embedding vectors across layers in encoder-only transformers without MLPs. The authors show that, under the hard-max rule, these dynamics can be viewed as a Frank–Wolfe step for a quadratic objective over the convex hull of the current token embeddings. Using this, the authors are able to characterize the dynamics of tokens across self-attention layers. When the key–query matrix is negative definite, it reinforces clustering behavior of the tokens, whereas when the key–query matrix is positive definite, a stable Voronoi structure is created.

All the reviewers and I agree that this is a very interesting paper. Despite studying a toy model, the results provide interesting insights into the across-layer dynamics of the attention mechanism.